

# SEAS5: The new ECMWF seasonal forecast system

Stephanie J. Johnson[1], Timothy N. Stockdale[1], Laura Ferranti[1], Magdalena Alonso Balmaseda[1],
Franco Molteni[1], Linus Magnusson[1], Steffen Tietsche[1], Damien Decremer[1], Antje Weisheimer[1],
Gianpaolo Balsamo[1], Sarah Keeley[1], Kristian Mogensen[1], Hao Zuo[1], and Beatriz Monge-Sanz[1]

[1]ECMWF, Shinfield Park, Reading RG2 9AX

*Correspondence to:* Stephanie J. Johnson (s.johnson@ecmwf.int)

**Abstract.**

In this paper we describe SEAS5, ECMWF's fifth generation seasonal forecast system, which became operational in November 2017. Compared to its predecessor, System 4, SEAS5 is a substantially changed forecast system. It includes upgraded versions of the atmosphere and ocean models at higher resolutions, and adds a prognostic sea ice model. Here, we describe

the configuration of SEAS5 and summarise the most noticeable results from a set of diagnostics including biases, variability, teleconnections and forecast skill.

An important improvement in SEAS5 is the reduction of the Equatorial Pacific cold tongue bias, which is accompanied by a more realistic ENSO amplitude and an improvement in ENSO prediction skill over the central-west Pacific. Improvements in two-metre temperature skill are also clear over the tropical Pacific. SST biases in the northern extratropics change due to

increased ocean resolution, especially in regions associated with western boundary currents. The increased ocean resolution exposes a new problem in the northwest Atlantic, where SEAS5 fails to capture decadal variability of the North Atlantic subpolar gyre, resulting in a degradation of DJF two-metre temperature prediction skill in this region. The prognostic sea ice model improves seasonal predictions of sea ice cover, although some regions and seasons suffer from biases introduced by employing a fully dynamical model rather than the simple, empirical scheme used in System 4. There are also improvements

in two-metre temperature skill in the vicinity of the Arctic sea-ice edge. Cold temperature biases in the troposphere improve, but increase at the tropopause. Biases in the extratropical jets are larger than in System 4: extratropical jets are too strong, and displaced northwards in summer. In summary, development and added complexity since System 4 has ensured SEAS5 is a state-of-the-art seasonal forecast system which continues to display a particular strength in ENSO prediction.

**1 Introduction**

The European Centre for Medium-Range Weather Forecasts (ECMWF) has been running real-time seasonal forecast systems since 1997. The seasonal system has been upgraded at approximately five year intervals during this time. SEAS5, ECMWF's



fifth generation seasonal forecast system, became operational in November 2017, replacing its predecessor "System 4" (hereafter SEAS4, Molteni et al., 2011) which had been operational since 2011.

SEAS4 was a state of the art seasonal forecast system, which maintained competitive performance over the six years it was operational. One particular feature was high ENSO forecast skill (Molteni et al., 2011). It also displayed good performance in

the prediction of the stratosphere and quasi-biennial oscillation (QBO) (e.g. Scaife et al., 2014). As with many other seasonal forecast systems, mid-latitude skill remained limited, although some skill was demonstrated in predicting southern European summer temperatures (Molteni et al., 2011) and sign of the Arctic Oscillation in northern hemisphere winter (Stockdale et al., 2015). Measures of overall skill in SEAS4 showed progress over previous systems (Molteni et al., 2011; Weisheimer and Palmer, 2014).

SEAS5 benefits from recent developments in its component models and initial condition generation. The IFS atmosphere model has improved since SEAS4 was implemented, especially in the representation of tropical convection (e.g. Bechtold et al., 2014), and there has been a substantial increase in horizontal resolution. The ocean model has also been upgraded with improved physics, increased horizontal and vertical resolution and a corresponding ocean and sea-ice reanalysis with up-to-date reprocessed observational data sets. SEAS4 lacked a prognostic sea-ice model, which is now considered an important

ingredient for seasonal forecasting, and has been included in SEAS5.

The benefits and challenges of a seamless forecasting system have been well documented in the literature (e.g. Brown et al., 2012). Development of a new seasonal forecast model at ECMWF has always used a recent version of the medium-range weather forecast model, with components added as needed to allow forecasting of longer timescales. Some of the components originally developed for the seasonal forecast system have later been adopted in the medium-range forecast model, most

notably an initialised ocean model (Janssen et al., 2013) and, consequently, the fundamental differences between the seasonal and medium-range forecast configurations have reduced over time. This trend has continued with the introduction of SEAS5. The starting point for SEAS5 was the forecast model configuration used in the ENS-extended ensemble forecast, which is targeted at forecasting the time range of 10 to 46 days. A few changes that were demonstrated to improve seasonal forecast skill were made to create the final SEAS5 configuration. Some of these changes have already been adopted by subsequent

versions of the medium-range forecast systems, in other cases the convergence is planned for the future.

The purpose of this paper is to document SEAS5 and outline its strengths and weaknesses compared to its predecessor, SEAS4. Given the very large number of metrics, scores, processes, geographical regions and modes of variability that we assess when introducing a new system, it is not feasible to document all all of them or expect that every single aspect of forecast performance is improved. However, it is important to present metrics that summarise performance and illustrate any

changes in the characteristics of the forecast system. In Section 2, we describe SEAS5 including the forecast and reforecast production (Section 2.1), the atmosphere and ocean model configurations (Section 2.2) and initial conditions for atmosphere and ocean (Section 2.3). Section 3 discusses the scope of our assessment and the statistical methods used in this paper. Section 4 uses diagnostics to describe the SEAS5's mean state climatology and the interannual variability of processes such as ENSO. Section 5 presents verification of the global performance of the system. We summarise the results in Section 6.

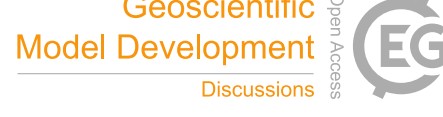



## 2 Description of SEAS5

### 2.1 Reforecast and forecast production

The "long range" forecast consists of a 51-member ensemble initialised every month on the first day of the month (see Section 2.3), and integrated for seven months. On each 1 February, 1 May, 1 August and 1 November, 15 of the 51 ensemble members are extended a further six months for a total forecast length of 13 months. These "annual range" forecasts were designed primarily to give an outlook for ENSO.

To verify the system and calibrate the forecast, SEAS5 uses a set of retrospective seasonal forecasts for past dates that can be compared to the historical record. This set of re-forecasts (also sometimes known as hindcasts) start on the first of every month for years 1981 to 2016 and have 25 ensemble members. This is a substantial increase on the SEAS4 operational re-forecast set which included 15 members initialised from 1981 to 2010. On 1 February, 1 May, 1 August and 1 November, 15 of the 25 SEAS5 re-forecast ensemble members are extended a further 6 months to provide a re-forecast set for the annual range forecasts. The entire re-forecast set is used to verify the forecast system (see Section 3), but only a subset of this re-forecast data, from years 1993 to 2016, is used in the calculation of forecast anomalies. Using this more recent period avoids the long term trend of climate change from overly affecting the forecast products, and also coincides with the calibration period used in the Copernicus Climate Change Service's multi-system seasonal forecast. SEAS5 became operational at the beginning of November 2017. In addition to the re-forecast set, 51 member forecasts were computed for all start dates in 2017 to allow assessment of SEAS5 on any initialisation date from 1 January 1981 to the current date.

### 2.2 Model configuration

Table 1 summarises the configuration of SEAS5 and compares it to SEAS4. SEAS5 uses updated versions of the atmosphere and ocean models and adds a new interactive sea ice model, and each of these components are described in detail below.

#### 2.2.1 Atmosphere model and forcing

SEAS5 uses ECMWF's Integrated Forecast System (IFS) atmosphere model cycle 43r1. A brief description of the parametrisations in the IFS is provided here, and the most significant changes between IFS cycle 36r4 (SEAS4) and 43r1 (SEAS5) are highlighted.

The radiation code is based on the Rapid Radiation Transfer Model (RRTM, Mlawer et al., 1997; Iacono et al., 2008). Cloud-radiation interactions are taken into account using the McICA (Monte Carlo Independent Column Approximation) method (Morcrette et al., 2008). For computational efficiency, the radiation calculations are only called every 3 hours which gives a poor representation of the diurnal cycle. In cycle 43r1, this is mitigated by approximate updating at higher time frequency, reducing biases in stratospheric temperature and errors in the diurnal cycle of near-surface temperature (Hogan and Bozzo, 2015; Hogan and Hirahara, 2016).





|  | SEAS4 | SEAS5 |
|---|---|---|
| IFS Cycle | 36r4 | 43r1 |
| IFS horizontal resolution (dynamics) | T255 | T319 |
| IFS horizontal grid | Linear | Cubic octahedral |
| IFS horizontal resolution (physics) | N128 (80 km) | O320 (36km) |
| IFS vertical resolution (TOA) | L91 (0.01 hPa) | L91 (0.01 hPa) |
| IFS model stochastic physics | 3-scale SPPT and SPBS | 3-scale SPPT and SPBS |
| Coupling | OASIS3 | Single executable |
| Ocean model | NEMO v3.0 | NEMO v3.4.1 |
| Ocean horizontal resolution | ORCA 1.0 | ORCA 0.25 |
| Ocean vertical resolution | L42 | L75 |
| Sea ice model | Sampled climatology | LIM2 |
| Wave model resolution | 1.0 | 0.5 |

**Table 1.** Table comparing the configuration of SEAS4 and SEAS5. Abbreviations are defined in the text.

The parametrisation of convection is based on the mass-flux approach (Tiedtke, 1989; Bechtold et al., 2008). The convective parametrisation evolves with each cycle, and in SEAS5 it has a modified CAPE closure leading to an improved diurnal cycle of convection (Bechtold et al., 2014) and a revised formulation of detrainment and convective momentum transport improving the tropical flow. The cloud and large-scale precipitation scheme (Tiedtke, 1993; Forbes et al., 2011; Forbes and Tompkins, 2011)

has an improved representation of mixed-phase clouds in cycle 43r1 (Forbes and Ahlgrimm, 2014). In addition, there were numerous other improvements to the parametrisation of microphysics, particularly for warm-rain processes (Ahlgrimm and Forbes, 2014), but also ice-phase processes and ice supersaturation. The combination of changes in the cloud and convection schemes between SEAS4 and SEAS5 substantially reduces biases in tropical temperature throughout the troposphere, as will be seen in Section 4.2.

The orographic gravity wave drag is parametrised following Lott and Miller (1997); Beljaars et al. (2004) and the non-orographic gravity wave drag parametrisation is as described in Orr et al. (2010). The turbulent mixing scheme follows the Eddy-Diffusivity Mass-Flux (EDMF) framework, with a K-diffusion turbulence closure and a mass-flux component to represent the non-local eddy fluxes in unstable boundary layers (Köhler et al., 2011). In cycle 43r1, the degree of turbulent mixing in stable conditions has been reduced to improve the representation of low level jets. This change combined with an increase in the

orographic drag led to significantly better representation of the large-scale circulation (Sandu et al., 2014). The representation of near surface winds was also improved by a revision of the roughness length (Sandu et al., 2011).

The surface-exchange parametrisation is based on a tiled approach (HTESSEL, Viterbo and Beljaars, 1995; Van den Hurk et al., 2000; Balsamo et al., 2009; Dutra et al., 2010a; Boussetta et al., 2013) representing different sub-grid surface types for vegetation, bare soil, snow and open water. The hydrology for soil infiltration and run-off is described by Balsamo et al. (2009)

and the representation of surface snow is described in Dutra et al. (2010a). For cycle 43r1, a representation of inland-water





bodies that can carry significant thermal storage and anomalies in the forecasts has been introduced (Mironov et al., 2010; Dutra et al., 2010b; Balsamo et al., 2012). In Cycle 43r1, the skin temperature for ocean points takes account of the cool skin effect and a diurnal warm layer effect (Zeng and Beljaars, 2005).

SEAS5 was developed following a "seamless" approach, so the atmospheric component of SEAS5 is nearly identical to the

IFS cycle 43r1 configuration used for the ENS extended forecast (IFS, 2016), which was operational for medium- and extended-range forecasting from 22 November 2016 to 11 July 2017. The atmospheric model uses a two-time-level semi-Lagrangian scheme, with spectral horizontal resolution of T319 and a 20 minute time-step. The model physical parametrisations are calculated in physical space on a reduced O320 Gaussian grid, which has grid spacing of approximately 36 km. There are 91 levels in the vertical, with a model top in the mesosphere at 0.01 hPa, or around 80 km. The ECMWF wave model is used at 0.5°

resolution (IFS, 2016, Part VII) with the same time-step as the atmosphere. One change to the cycle 43r1 model settings was introduced for SEAS5. In SEAS5 the tropical amplitude of the non-orographic gravity wave drag was considerably reduced compared to the default settings in 43r1 in order to improve the modelling of the QBO and the climate mean stratospheric winds. The impact of this change is described in Section 4.4.

Greenhouse gas radiative forcing is taken from CMIP5. A new prognostic ozone scheme (Monge-Sanz et al., 2011) replaces

the scheme used in SEAS4 and the default 43r1 configuration (Cariolle and Déqué, 1986; Cariolle and Teyssèdre, 2007), but as part of the seamless strategy used to develop SEAS5 prognostic ozone is not radiatively interactive as it was in SEAS4. Instead, the radiation scheme sees the same ozone climatology used in the cycle 43r1 ENS extended forecasts. Tropospheric sulphate aerosol follows the decadally-varying CMIP5 climatology, rather than the time invariant climatology that is default in cycle 43r1. Volcanic stratospheric sulphate aerosol is still treated by the method used for SEAS4; the initial load of volcanic

aerosol is prescribed using GISS data (2012 update[1]). The forecast is initialised using the GISS values from the month before the forecast starts, and then evolved in time with damped persistence (timescale 400 days). The vertical distribution follows a prescribed profile that is dependent on the depth of the stratosphere. The horizontal distribution is approximated by three numbers, the northern hemisphere, tropical and southern hemisphere amounts. SEAS5 cannot predict volcanic eruptions, but after a major eruption occurs, manual estimates of the volcanic aerosol, based in part on Copernicus Atmosphere Monitoring

Service (CAMS) $SO_2$ analyses, could be included in future real-time forecasts. The new prognostic ozone scheme is used to determine the tropopause height for application of volcanic aerosol.

### 2.2.2 Ocean and cryosphere models

SEAS5 uses the Nucleus for European Modelling of the Ocean model (NEMO, Madec, 2016) version 3.4.1 developed by the NEMO European consortium, which is an upgrade from the NEMO v3.0 model used in SEAS4. It contains upgrades to aspects

of ocean-surface wave interaction (Breivik et al., 2015) originally introduced at ECMWF, including estimating momentum flux from the dissipation term (accounting for the intensity of breaking waves); accounting for the energy flux from breaking waves in surface boundary condition of the turbulent kinetic energy equation (Craig and Banner, 1994); and introducing the Coriolis-Stokes forcing term in the momentum equation.

---

[1]https://data.giss.nasa.gov/modelforce/strataer/



The ocean model horizontal resolution increases from ORCA1° in SEAS4 to ORCA0.25° (developed by the DRAKKAR international research network) in SEAS5, which improves the representation of sharp fronts and ocean transports in SEAS5. The number of ocean vertical levels increases from 42 to 75, including an increase from 5 to 18 levels in the uppermost 50 metres of the ocean. This reduces the depth of the surface layer of the ocean model from 10-metres to 1-metre, which improves

the representation of the diurnal cycle of sea-surface temperatures (SST). The ocean model time-step is 20 minutes.

The Louvain-la-Neuve sea ice model version 2 (LIM2, Fichefet and Maqueda, 1997), developed at the Belgian Université catholique de Louvain, is added in SEAS5. Introducing a prognostic sea-ice model allows the sea-ice cover to respond to changes in the atmosphere and ocean states, enabling SEAS5 to provide seasonal outlooks of sea-ice cover. At the same time, prognostic sea ice has the potential to improve forecasts of the atmospheric state and circulation by virtue of improved

surface fluxes of heat, moisture and momentum. LIM2 is part of the NEMO modelling framework and uses the same tripolar ORCA0.25° grid as the ocean, but has an hourly time step. It is a dynamic-thermodynamic model, with a single thickness category. The model is used within SEAS5 to simulate the evolution of the fractional ice cover (sea-ice concentration), and only this variable is coupled to the atmosphere surface scheme. LIM2 simulates the conductive heat flux within the ice based on two vertical layers in the ice with varying thickness and a single snow layer on top of the ice, which determines the basal

ice growth rate during winter. The surface heat flux at the sea-ice-atmosphere interface, however, is determined by an ice conductive heat flux computed by the atmosphere model. This leads to thermodynamic inconsistencies at the surface, resulting in an overestimation of the basal ice growth rate in winter, as seen in Section 4.3. The model also does not simulate the formation or evolution of melt ponds which is important for summer surface energy balance. Ice velocities are computed by solving an appropriate momentum balance equation using a viscous-plastic rheology; sea-ice velocities are important because

they give rise to the transport of sea-ice properties by advection.

### 2.2.3   Coupling

Some of the model components are tightly coupled: the land component, being on the same grid as the atmosphere model and requiring only vertical physics, has always been embedded within the atmosphere model; the ocean and sea-ice components are also tightly coupled to each other. A coupling interface then computes exchanges of information between three distinct modules

that use three different horizontal grids: the atmosphere-land, the ocean and sea-ice, and the wave model. The atmosphere and wave models exchange fluxes of heat, momentum, freshwater and turbulent kinetic energy with the ocean and sea-ice, while the ocean and sea ice models communicate SST, surface currents and sea ice concentration to the atmosphere and wave models. There is no coupling between land and ocean.

The coupling interface in SEAS5 is implemented as a single executable where as SEAS4 used the OASIS3 coupler (Valcke,

2006). Details on the single executable coupling interface can be found in Mogensen et al. (2012). As in SEAS4 (Molteni et al., 2011), a Gaussian method is used for interpolation between the atmosphere and ocean models in both directions, primarily due to the complexity of the 0.25° ORCA grid. The Gaussian method automatically accounts for the different coast lines of the atmosphere and ocean models - values at land points are never used in the coupling, since these can be physically very different to conditions over water. The atmosphere and ocean are coupled hourly to allow the diurnal cycle to be resolved.



|  | SEAS4 | SEAS5 |
|---|---|---|
|  | re-forecast/forecast | re-forecast/forecast |
| Atmosphere initialisation | ERA-Interim/Operations | ERA-Interim/Operations |
| Land Initialisation | ERA-Interim land (36r4)/Operations | ERA-Interim land (43r1)/Operations |
| Ocean initialisation | ORA-S4/ORTA4 | ORA-S5/OCEAN5-RT |

**Table 2.** Table summarising the initialisation of SEAS4 and SEAS5. Abbreviations are defined in the text.

### 2.3 Model initialisation

Table 2 summarises the main datasets used to initialise SEAS5 and compares them to SEAS4. The model used to calculate SEAS5 forecasts and re-forecasts is identical, but forecasts must be initialised differently from re-forecasts in order to make use of near-real time observational data. Forecasts and re-forecasts should be initialised and calculated as similarly as possible to ensure accurate bias correction. We describe the initialisation of both re-forecasts and forecasts here, including any adjustments made to improve consistency between re-forecast and forecast initialisation.

#### 2.3.1 Atmosphere and Land

In SEAS5 re-forecasts (prior to 1 January 2017) the atmosphere is initialised from ERA-Interim (Dee et al., 2011). Forecasts (1 January 2017 and later) are initialised from ECMWF operational analyses.

The inter-annual variability of ozone in ERA-Interim is affected by changes in satellite instruments over time, and does not represent the true inter-annual variability of ozone in the atmosphere (Dee et al., 2011). Consequently, the prognostic ozone scheme is instead initialised with a seasonally varying climatology produced by the ozone model (Monge-Sanz et al., 2011) within an integration where an enhanced vertical resolution version of the IFS (cycle 42r1, L137) is nudged to ERA-Interim vorticity (12h timescale) and tropopause temperature (5 day timescale, which is needed to control biases in lower stratosphere temperature).

Land surface initial conditions for the re-forecasts are generated by the Cycle 43r1 version of the HTESSEL scheme run in offline mode for the re-forecast period at the same resolution as SEAS5. In offline mode, HTESSEL is forced with ERA-Interim (precipitation, solar radiation, near surface temperature, winds and humidity) following the method described in Balsamo et al. (2015).

The land surface in SEAS5 forecasts is initialised from ECMWF operational analysis, which includes a dedicated land data assimilation as described in de Rosnay et al. (2014). The SEAS5 land initial conditions are then interpolated from the HRES O1280 grid onto the O320 SEAS5 grid. This interpolation can result in locally large differences compared to initial conditions prepared directly at the lower resolution. Consequently, a limiter is used to prevent the real-time land surface values taking inconsistent values relative to those used in the re-forecasts. The limits are defined as the maximum and minimum values





observed at that point and calendar date for the 36 year re-forecast period, plus an additional margin specified as a global constant for each field. For more details please refer to the SEAS5 user guide[2].

### 2.3.2 Ocean

SEAS5 ocean and sea-ice initial conditions for forecasts and re-forecasts are provided by the new operational ocean analysis

system, OCEAN5 (Zuo et al., 2018), which is made up of the historical ocean reanalysis (ORAS5) and the daily real time ocean analysis (OCEAN5-RT). OCEAN5 uses the same ocean and sea-ice model as the coupled forecasts in SEAS5. OCEAN5 is conducted with NEMOVAR (Mogensen et al., 2012) in its 3D-Var FGAT (First-Guess at Appropriate Time) configuration. Compared to its predecessor ORAS4 (Balmaseda et al., 2013), OCEAN5 has higher resolution, updated data assimilation and observational data sets and provides sea-ice initial conditions.

ORAS5 is based on Ocean Reanalysis Pilot 5 (ORAP5, see Zuo et al., 2017b; Tietsche et al., 2017), but using updated observational data sets. The ocean in-situ temperature and salinity comes from the recent quality-controlled EN4 (Good et al., 2013), which has higher vertical resolution and fuller spatial coverage than the previous version EN3. The altimeter sea-level data have also been updated to the latest version (AVISO DT2014, Pujol et al., 2016) from CMEMS (Copernicus Marine Environmental Monitoring Services). The underlying SST analysis before 2008 comes from the HadISST2 dataset (Titchner

and Rayner, 2014). The sea-ice concentration before 1985 comes from ERA-40 and from 1985 to 2008 it comes from an OSTIA (Donlon et al., 2012) reprocessed product. From 2008 onwards the SST and sea-ice are given by the OSTIA product delivered in real-time, which is also used in the ECMWF operational analysis. More details on the system configurations and sensitivities are detailed in Zuo et al. (2018).

## 2.4 Ensemble generation

### 2.4.1 Initial condition perturbations

Initial condition perturbations are applied to atmosphere and ocean initial conditions to represent uncertainty in the initial state and increase ensemble spread. Ensemble member 0 is initialised from unperturbed atmospheric initial conditions, in other members all upper air fields and a limited set of land fields (soil moisture, soil temperature, snow, sea-ice temperature and skin temperature) are perturbed. As in the operational ENS, perturbations from an ensemble of data assimilations (EDA) and

perturbations constructed from the leading singular vectors are applied (IFS, 2016, Part V). EDA perturbations are not available for the earlier years in the re-forecast set, so to preserve consistency across the hindcast set and forecasts, the EDA perturbations from 2015 were applied to the initial conditions for all forecast and re-forecast years.

OCEAN5 contains a 5-member ensemble analysis. The perturbation scheme used to generate the ensemble of re-analyses consists of two distinct elements: perturbations to the assimilated observations, both at the surface and at depth, and pertur-

bations to the surface forcing fields (Zuo et al., 2017a). The ocean analysis near-surface temperatures are further perturbed so that all ensemble members are initialised from slightly different ocean initial conditions. These perturbations are drawn from

---

[2]https://www.ecmwf.int/sites/default/files/medialibrary/2017-10/System5_guide.pdf





the ORAS5 HadISST2 pentad analysis error repository (Zuo et al., 2017a, Section 4), and applied to the upper 22 levels of the sea temperature, decreasing with depth. This perturbation is not applied to ensemble member 0.

### 2.4.2 Stochastic model perturbations

In addition to perturbing the initial conditions, perturbations to the atmospheric model are also applied, to represent uncer-
tainty from missing or unresolved sub-grid scale processes (e.g. convection, clouds, radiation, turbulence) which have to be parameterised (Palmer, 2012). ECMWF has been using stochastic parametrisation schemes to explicitly account for these uncertainties in its forecasting systems from the medium-range to seasonal for many years (Buizza et al., 1999; Palmer et al., 2009) and the schemes that are used in SEAS5 are identical to those used in the shorter forecast ranges (see Leutbecher et al., 2017). The Stochastically Perturbed Physical Tendency (SPPT) scheme introduces flow-dependent multiplicative noise to the
total tendencies of the prognostic variables temperature, horizontal wind and humidity at model levels. The noise has a spatial and temporal correlation structure with three distinct scales representing small-scale fast perturbations, large-scale slow perturbations and an intermediate scale. A tapering in the boundary layer and the upper-most model levels effectively switches off the SPPT perturbations in these regions. The version of SPPT used here is based on a mass, energy and moisture conservation fix that was originally developed by the EC-Earth consortium, see Lang et al. (2016). The Stochastic Kinetic Energy Backscatter
(SKEB) scheme aims at improving the upscale energy cascade from the sub-grid scales to the resolved scales (Shutts, 2005), but has been found to have a smaller overall impact in the ECMWF system (Weisheimer et al., 2014). For details of the schemes and performances, see Palmer et al. (2009); Lang et al. (2016); Leutbecher et al. (2017) and Weisheimer et al. (2014). Stochastic perturbations from both SPPT and SKEB are applied to all ensemble members: SEAS5 does not have a control forecast.

## 3  Assessment scope and evaluation methods

In order to compare the SEAS5 skill with the previous operational system (SEAS4), we could work with the largest common
period for which the re-forecasts from SEAS4 and SEAS5 are available, namely 1981 to 2010. Since a key component of the seasonal forecast skill is the ability to forecast ENSO, it is important to consider a long verification period to include sufficient numbers of ENSO events. To allow a longer verification period we have included the operational forecasts for SEAS4 for 2011 to 2016, giving an overall comparison period of 1981 to 2016. This choice is not perfect since there are inconsistencies in the land surface initialisation between the SEAS4 re-forecasts and SEAS4 real-time forecasts. Comparison of the SEAS5
and SEAS4 score differences for 1981 to 2010 and 1981 to 2016 (not discussed in this paper) shows no sign of this slight inconsistency affecting the results presented here. Consequently, the assessment in this paper is based on this 36 year re-forecast period unless otherwise mentioned (see Section 3.2), which is consistent with the SEAS5 verification available on ECMWF's website[3].

SEAS5 has an increased operational re-forecast ensemble size compared to SEAS4, however the real-time ensemble size
is the same in both systems. Since we are interested in the comparative skill of the real-time forecast system, throughout this

---

[3]https://www.ecmwf.int/en/forecasts/charts/catalogue/?facets=Type,Verification%3BRange,Long%20(Months)



article we compare the two forecast systems using the same ensemble size. Since the implementation of SEAS4, extra ensemble members have been added to quarterly re-forecast dates (Feb, May, Aug, Nov), allowing us to compare the 25 member SEAS5 re-forecast set to 25 ensemble members from SEAS4. When only 15 SEAS4 ensemble members are available, we compare them to the first 15 members from SEAS5.

Our assessment is performed on monthly means. "Forecast lead time" is defined here to be the months elapsed since forecast initialisation but prior to the month being discussed, while "forecast month" includes the the month being discussed, one more than forecast lead. For example, if a forecast is initialised on the 1st of January, February has one month forecast lead time and is month two of the forecast. "Verification month" is defined as the calendar month that the forecast is issued for. Unless otherwise mentioned, diagnostics are seasonal means at one-month lead time (i.e. a DJF SST map is from a 1 November start

date), which corresponds to months two to four of the forcast.

### 3.1 Evaluation and verification metrics

The seasonal forecast performance has been evaluated using a wide range of deterministic and probabilistic scores. For ENSO forecasts and other SST statistics we use deterministic metrics such as anomaly correlation and root mean square errors. For the skill of atmospheric variables we also use probabilistic metrics such as the continuous ranked probability score and reliability

diagrams.

#### 3.1.1 Anomaly correlation

Anomaly correlation is calculated in two ways in this paper. For tropical ocean (e.g. ENSO) indices, QBO indices and deterministic skill maps estimates (Figures 19 and 20), anomaly correlation is calculated in accordance with established practice for scoring ENSO forecasts. First, bias-corrected anomalies for each forecast date and lead-time in the re-forecast dataset,

are created using cross-validation (i.e., the bias correction is calculated only from other re-forecast years, not the one being bias corrected). Anomalies for tropical ocean indices are calculated with respect to a standard 30 year reference climate period, which is 1981 to 2010. Anomalies for deterministic skill maps are calculated with reference to the full validation period of 1981 to 2016. The correlation is then calculated between the ensemble mean forecast and observed anomaly time-series. The cross-validation procedure affects the correlation negatively, leading in theory to a small but systematic underestimate of

expected future forecast skill.

For all other indices, no cross-validation is applied and the anomalies are computed with respect to the entire time-series, 1981 to 2016.

#### 3.1.2 Amplitude ratio

The ratio of the forecast anomaly amplitudes to observed amplitudes is calculated from the cross-validated bias corrected indi-

vidual ensemble member anomalies, computed with respect to 1981 to 2010. The standard deviation of the forecast anomalies





is calculated from the mean square amplitude of all ensemble members and all start years (for a given start month and lead time), and then compared with the standard deviation of observations.

### 3.1.3   Root mean square error

For tropical ocean and QBO indices, root mean square error (RMSE) of tropical ocean indices is calculated from the cross
validated bias-corrected ensemble mean of the forecasts. For all other indices, cross validation is not applied.

### 3.1.4   CRPSS

The CRPSS (Wilks, 2011) is calculated for each variable's seasonal average at each grid point for each year of the whole re-forecast period. If follows that the CRPSS map is estimated over 36 independent events. The climatology computed over the 36 years re-forecast period is used as the reference forecast. Therefore the CRPSS gives an indication of the added value
of a forecasting system over simply forecasting climatology, a value of one indicating perfect forecasts, zero showing no improvement over climatology and negative values indicating a failing forecasting system. Significance-testing for the CRPSS differences between SEAS5 and SEAS4 is evaluated at a 5%-significance level with a Z-test on pairwise bootstrapped CRPSS differences. For this Gaussian-approximated bootstrap method, we resample the forecasts and ensemble members over 1000 repetitions (with replacement) to capture the uncertainty both in time and in the ensemble.

### 15   3.1.5   Reliability

Reliability diagrams are used to summarise whether the forecast probabilities agree with the observed frequency of occurrence of a binary event (e.g. temperature in the upper tercile). To create the reliability diagrams used in this paper, each forecast at every grid point within a selected region is binned into one of 26 bins (one more than the number of ensemble members) according to the forecasted likelihood of occurrence of the chosen event. This likelihood is then plotted against the frequency
with which the event actually occurred for this sub-set of forecasts and grid-points. In a perfectly reliable system, the forecast probability will equal the frequency of occurrence and the values for each bin will lie along a straight diagonal line in the reliability diagram. Uncertainties are computed by bootstrapped resampling over years and ensemble members.

### 3.2   Datasets

For most variables the ERA-Interim re-analysis was used for verification (Dee et al., 2011), which is also the atmosphere
initialisation data for SEAS4 and SEAS5. To verify precipitation we use Global Precipitation Climatology Project (GPCP) Monthly Precipitation Analysis 2.2 (Adler et al., 2003). Since GPCP 2.2 data is not available for the whole re-forecast period precipitation verification statistics are based on 1981 to 2014 period.

The depth of the surface layer of the ocean model decreases from 10-metres in SEAS4 to 1-metre in SEAS5, which changes the depth that SST is calculated from. To ameliorate the impact of this difference on the SST biases, we initially compare SST
maps in each system to the analysis it was initialised from, ORAS4 (Balmaseda et al., 2013) or ORAS5 (Zuo et al., 2018). Later,



area-averaged SST indices are compared to the OI.v2 reanalysis (OIv2, Reynolds et al., 2002) or ERA-Interim re-analysis, to measure both systems against the same standard. As will be seen in Section 4, when averaging over large regions, consistent conclusions are reached regardless of observational dataset used.

ERA-Interim sea ice is not temporally consistent, and is not recommended as a sea ice verification data set. Instead, we use the EUMETSAT Ocean and Sea Ice Satellite Application Facilities' (OSI SAF) global sea ice concentration climate data record (OSI-450)[4]. OSI-450 is the second major version of the OSI SAF Global Sea Ice Concentration Climate Data Record. The sea ice concentration is computed from the SMMR (1979-1987), SSM/I (1987-2008), and SSMIS (2006-2015) instruments. The OSI-450 product is available from 1979 to 2015, but because of gaps in the satellite record, data is not available for every day. We have taken the choice that if five consecutive days of data are missing from any season, that season is left out of our evaluation of sea ice concentration. Consequently, in JJA we exclude 1984, 1986, 2016; in DJF we exclude 1986, 1987, 1990, 2015, 2016, in MAM we exclude 1981, 1986 and 2016 and in SON we exclude 2016.

## 4 SEAS5 diagnostics: Climate and interannual processes

In this section we use diagnostics of interannual processes to assess SEAS5 and compare it to SEAS4. We first discuss the tropics, with a focus on tropical SST variability (Section 4.1). Then we discuss the northern extratropics, with a particular focus on the North Atlantic SST (Section 4.2). Finally we discuss the impact of introducing the prognostic sea ice model LIM2 (Section 4.3) and the representation of the stratosphere (Section 4.4), before going on to discuss the global verification of SEAS5 in the next section.

### 4.1 Tropics

Interannual modes of variability in tropical oceans are the primary known source of seasonal predictability (e.g. Charney and Shukla, 1981; Palmer and Anderson, 1994; Stockdale et al., 1998; Troccoli, 2010). Consequently, a realistic representation of the tropical variability is a crucial requirement for a successful seasonal forecasting system. In Figure 1, we show the tropical SST bias in SEAS4 and SEAS5 relative to the ocean reanalysis they were initialised from, ORAS4 or ORAS5 (see Section 3.2). The tropical oceans are generally warmer in SEAS5, especially in the summer hemisphere. This overall warming is mainly due to changes in the ocean vertical mixing, which produces shallower mixed layers within the tropical regions. Warm biases flank the equator in the tropical Pacific and Atlantic basins. In the Indian Ocean and west Pacific, cold biases in SEAS4 are replaced with a warm bias in SEAS5. There is a reduction in the equatorial Pacific cold tongue bias in SEAS5 that exceeds 5°C at its maximum in SEAS4. Initial investigations suggest that the both the change in ocean model horizontal resolution and improvements in the atmosphere model contribute to the reduction of the cold tongue bias. Improvements in IFS tropical convection and cloud physics give higher total column water vapour in SEAS5, with more absorption of thermal radiation, resulting in a reduction in tropical outgoing long-wave radiation of 3.0 W m$^{-2}$ in DJF and 2.4 W m$^{-2}$ in JJA. This change to the atmosphere radiative balance may contribute to the changes in tropical SST.

---

[4]http://osisaf.met.no/docs/osisaf_cdop2_ss2_pum_sea-ice-conc-climate-data-record_v1p0.pdf





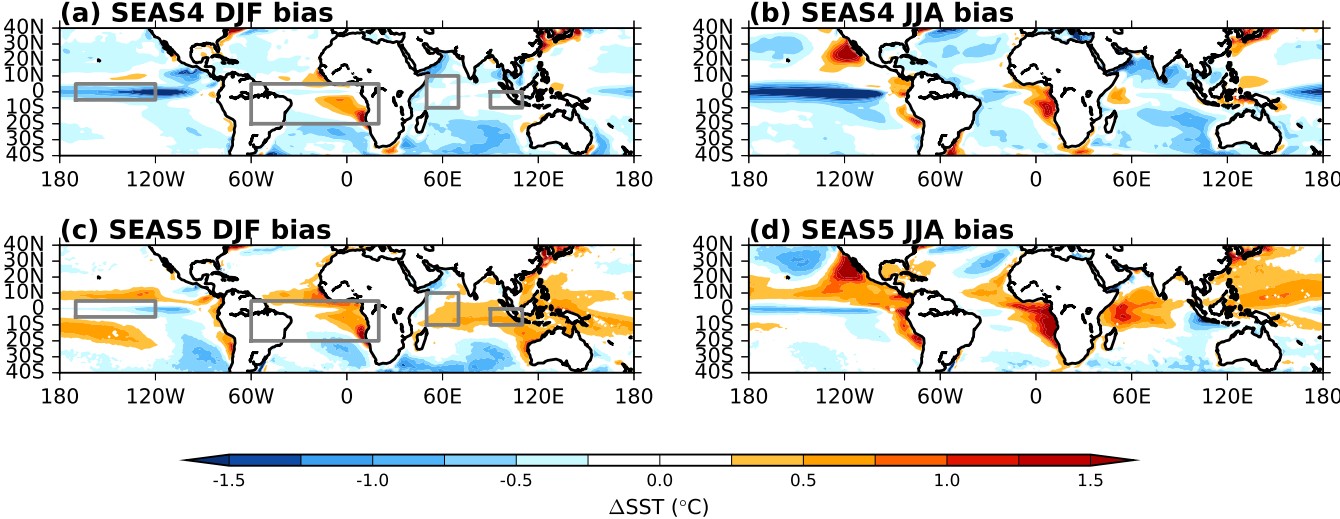

**Figure 1.** Winter and summer SST bias in the tropics at one month lead forecast for SEAS4 (a,b) and SEAS5 (c,d) compared to the analysis they were initialised from (ORAS4, ORAS5). The regions discussed in detail later in this section are outlined in grey here.

The dominant mode of global SST interannual variability is the El Niño Southern Oscillation (ENSO, e.g. McPhaden et al., 2006; Deser et al., 2010). Figure 2 shows the mean state bias, amplitude ratio, anomaly correlation and root mean square error (RMSE) of the Niño 3.4 region (-120° to -170°W, 5°N to 5°S, illustrated in Figure 1) as a function of forecast lead time using 15 ensemble members and all start dates from the SEAS4 and SEAS5 hindcast set. For a detailed description of these metrics

see Section 3.1. In order to compare the systems on an equal footing, these diagnostics are computed relative to the NCEP OIv2 (see Section 3.2). Both the long range forecast (solid lines) and annual range forecast (dashed lines) are shown for SEAS4 and SEAS5 (see Section 2.1).

The decrease in the equatorial Pacific cold tongue bias seen in Figure 1 is also clear in Figure 2, with an improvement of nearly two degrees in the SEAS5 Niño 3.4 bias after 13 months of model evolution. The SEAS5 bias does not change very much

after the first few months of the forecast, while the SEAS4 bias continues to grow through the early parts of the annual range forecast. The other metrics in Figure 2 reveal that the interannual variability of ENSO has also improved. For the seven-month duration of the long-range forecast in both SEAS4 and SEAS5, the amplitude of the variability exceeds that of the analysis indicating the model is over-active in the equatorial Pacific. This over-activity is reduced in SEAS5, with an approximately 10% improvement in the amplitude ratio in the long-range forecast. ECMWF already had high skill in forecasting ENSO

compared to other state-of-the-art seasonal forecast models, especially in the spring and summer months that are more difficult to forecast (Barnston et al., 2012; Molteni et al., 2011). This skill is improved in SEAS5, with an improved anomaly correlation at all lead times, but particularly in the annual range forecast. These improvements combine to improve RMSE approximately by 0.1°C at forecast leads longer than one month. Improvements in ENSO skill are particularly noticeable in the Western-Central Pacific (e.g. Niño 4), while they are more modest in the Eastern part of the basin (e.g. Niño 3) (not shown). In spite

of these improvements, SEAS5 continues to be under-dispersive in the ENSO regions, the ensemble spread is approximately







**Figure 2.** Forecast performance metrics (Section 3.1) of the monthly averaged Niño 3.4 index in SEAS4 (blue) and SEAS5 (red). Long range (7 month) forecasts are shown as the solid lines, and use 15 ensemble members from each of the 12 monthly start dates. Annual range (13 month) forecasts are shown in the dashed lines and use 15 ensemble members from each of the four quarterly start dates. The verification data is NCEP OIv2. The top row shows a) climatological bias and b) ratio of the standard deviation of re-forecast and OIv2 anomalies, calculated using individual ensemble members. The bottom row shows c) anomaly correlation and d) RMSE.

80% of the RMSE at lead times longer than one month. SEAS5 is slightly more under-dispersive than SEAS4, due to a larger drop in the spread than improvement in the skill in the Niño3 region (not shown).

SEAS5 ENSO forecast skill, like ENSO anomalies themselves, varies throughout the year. In Figure 3 we show the same four

5   metrics that were depicted in Figure 2, but as a function of verification month. The solid lines are averaged over lead times





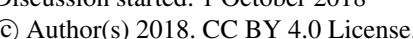

**Figure 3.** Forecast performance metrics of the monthly averaged Niño 3.4 index in SEAS4 (blue) and SEAS5 (red) as a function of verification month. The solid lines are averaged over one to three month lead times, and the dashed lines are averaged over four to six month lead times. Each line uses 15 ensemble members. The top row shows a) Niño 3.4 SST climatological bias and b) ratio of the standard deviation of the re-forecast and OIv2 anomalies, calculated using the individual ensemble members. The bottom row shows c) anomaly correlation and d) RMSE. The standard deviation of the interannual variability in OIv2, indicating the annual cycle of the interannual variability, is plotted as the dotted line in panel (d).

of one to three months, and the dashed lines are averaged over lead times of four to six months. Here we can see that while the Niño 3.4 bias improves throughout the year, it is particularly improved in late summer and autumn at longer lead times. In SEAS5, the bias is fairly consistent throughout the year, though it is a bit larger in spring at longer lead times. In both SEAS5 and SEAS4, the model is particularly overactive in spring, and this overactivity grows at longer lead times, whereas in the autumn and winter, overactivity diminishes at longer lead times. The anomaly correlation decreases at longer lead times, as expected, but particularly in the summer and early autumn. This is also seen in the annual range forecast. In Table 3 we show



| Forecast lead | January | | | July | | |
|---|---|---|---|---|---|---|
| | Forecast start date | SEAS5 | SEAS4 | Forecast start date | SEAS5 | SEAS4 |
| 2 | Nov | 0.98 | 0.97 | May | 0.88 | 0.86 |
| 5 | Aug | 0.93 | 0.88 | Feb | 0.71 | 0.72 |
| 8 | May | 0.89 | 0.76 | Nov | 0.52 | 0.60 |
| 11 | Feb | 0.78 | 0.59 | Aug | 0.52 | 0.58 |

**Table 3.** Niño 3.4 anomaly correlation values for the annual range forecast using the ensemble mean of 15 ensemble members, listed as a function of lead time (months) for two verifying months, January and July.

the anomaly correlation for Niño 3.4 January and July anomalies at forecast leads of 2, 5, 8 and 11 months. SEAS5 January skill is maintained well throughout the forecast, with an anomaly correlation of 0.78 at 12 months forecast lead. In SEAS4, this dropped to 0.59. This represents a considerable improvement in the ENSO prediction skill of the annual range forecast. In summer, when Niño 3.4 anomalies are generally smaller, the anomaly correlation in both systems has dropped below 0.6 by
eight months forecast lead and SEAS4 outperforms SEAS5 by a small margin.

Other important modes of tropical SST variability include the Indian Ocean dipole (IOD, Saji et al., 1999; Webster et al., 1999), and tropical Atlantic variability sometimes referred to as the Atlantic Niño (e.g. Zebiak, 1993). Figure 4 shows metrics as a function of verification month for the regions that form the IOD index: the western equatorial Indian ocean (WEIO, 50° to 70°E, 10°N to 10°S) and eastern equatorial Indian Ocean (EEIO, 90° to 110°E, 0° to 10°S). These regions are illustrated as
grey boxes on the maps in Figure 1. In the WEIO, a cold bias in SEAS4 becomes a warm bias in SEAS5, but the amplitude of the bias remains similar. Otherwise, very little changes from SEAS4 to SEAS5. The anomaly correlation shows some variation with season in both systems, with a particular drop in anomaly correlation mid-summer. In the EEIO, SEAS5 metrics degrade compared to SEAS4, and there is clear seasonality to this degradation. In an IOD event, a cold anomaly develops in the EEIO, off the coast of Sumatra. In SEAS5, these cold events develop most years, and with large amplitudes, likely related to a deficit
in precipitation in the EEIO and an easterly wind bias (not shown). The cold events are visible in Figure 4 b and d, where the bias is cold in the EEIO from July through autumn (depending on lead time) and the amplitude of the variability is much too large, nearly double the observed amplitude at longer lead times. This has a marked detrimental effect on the anomaly correlation and RMSE at longer lead times in late summer and early autumn.

In Figure 5 we show the bias and anomaly correlation for the equatorial Atlantic (-60°W to 20°E, 5°N to 20°S, region
illustrated in Figure 1). In the Atlantic, the SEAS5 bias is warmer and larger throughout the year compared to SEAS4. The amplitude of the interannual variability changes very little from SEAS4 to SEAS5 (not shown), but anomaly correlation increases slightly (Figure 5), leading to slight decreases in RMSE (not shown). In both systems, skill in the late summer, when Atlantic Niño variability peaks (Zebiak, 1993), is maintained through longer lead times, while it degrades at other times of the year.

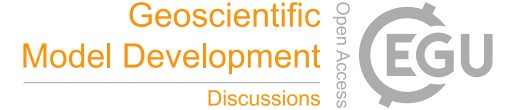



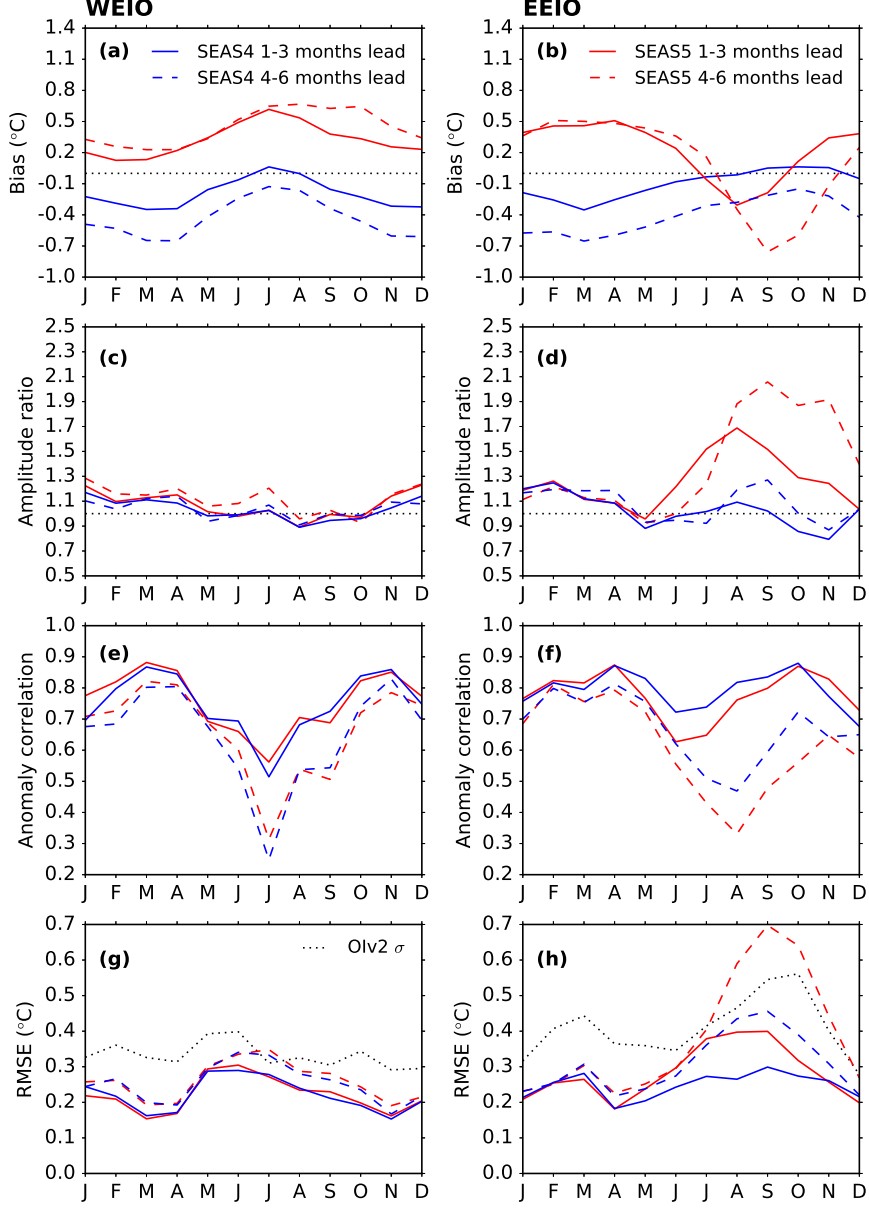

**Figure 4.** Forecast performance metrics of the regions that contribute to the Indian Ocean Dipole index (illustrated in Figure 1) as a function of verification month. a,b) SST climatological bias. c,d) ratio of the standard deviation of forecast and OIv2 anomalies. e,f) anomaly correlation. g,h) RMSE. SEAS4 is shown in blue and SEAS5 is shown in red. The solid lines are averaged over one to three month lead times, and the dashed lines are averaged over four to six month lead times. Each line uses 15 ensemble members. The standard deviation of the interannual variability in OIv2, indicating the annual cycle of the interannual variability, is plotted as the dotted line in panels g and h.





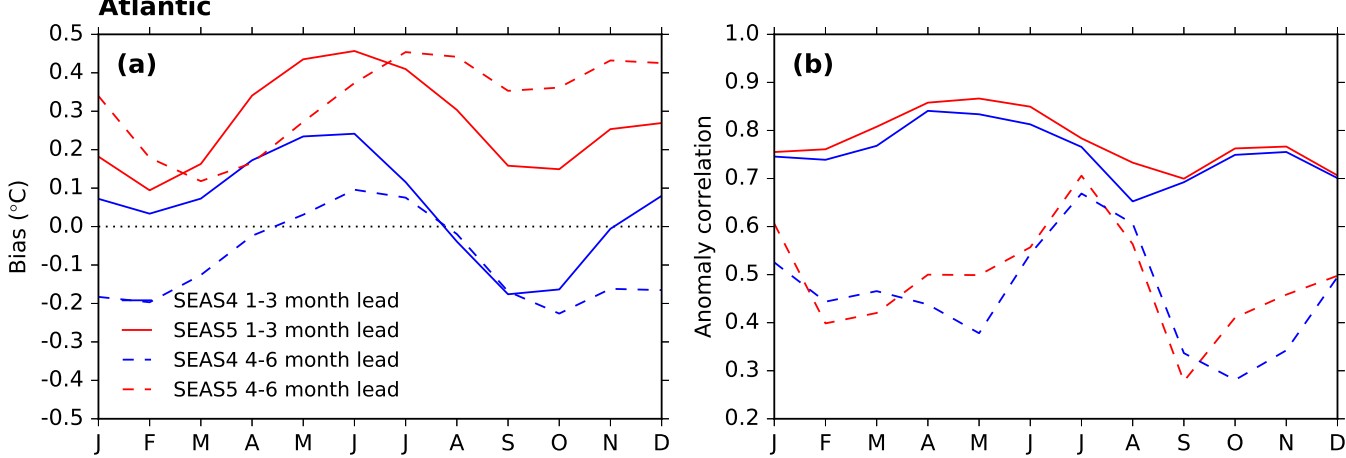

**Figure 5.** Forecast performance metrics for the tropical Atlantic region (illustrated in Figure 1) as a function of verification month. a) SST climatological bias and b) anomaly correlation. SEAS4 is shown in blue and SEAS5 is shown in red. The solid lines are averaged over one to three month lead times, and the dashed lines are averaged over four to six month lead times. Each line uses 15 ensemble members.

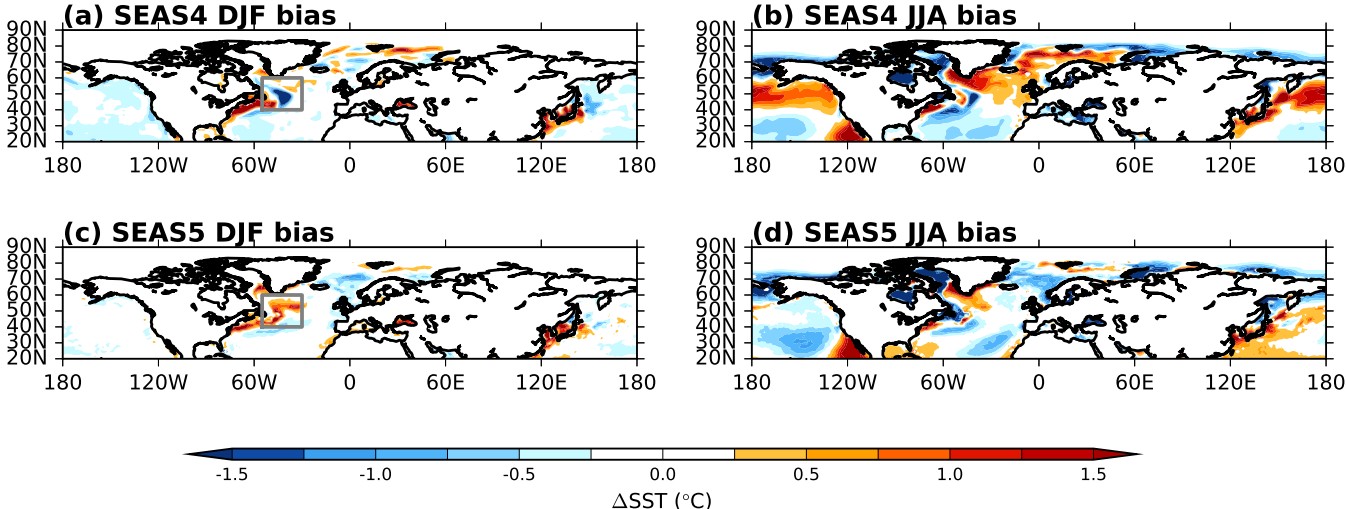

**Figure 6.** Summer and winter SST bias in the northern extratropics at one month forecast lead for SEAS4 (a,b) and SEAS5 (c,d) compared to the analysis they were initialised from (ORAS4, ORAS5). The region discussed in detail later in this section is outlined in grey here.

## 4.2 Northern extratropics

The SST bias in the northern extratropics is shown for both summer and winter in Figure 6. In the northern Pacific SST biases reduce, particularly in the summer. This is partly due to improved parametrisations for ocean vertical mixing. In the Northern Atlantic, increased horizontal resolution in the ocean model alters the path of the Gulf Stream, which changes SST biases. A

5    constant positive SST bias in the Gulf Stream region is present in both SEAS4 and SEAS5. This is connected to the long-





**Figure 7.** DJF time-series of SST anomaly in a northwest Atlantic region: 50 to 30°W, 40 to 60°N (illustrated as a grey box in Figure 6) at one month forecast lead. Quartiles, minimum and maximum of the SEAS4 25 member ensemble are shown in blue, while the SEAS5 25 member ensemble is shown in red. The black bars indicate ERA-Interim reanalysis. A five-year running mean for each system is shown as a dashed line. Forecasts were initialised in November, and the year shown is the year the ensemble was initialised.

standing and well-known failure of low resolution ocean models to simulate the separation of the Gulf Stream from the North American coast correctly (Chassignet and Marshall, 2008). This bias has improved in SEAS5. Further downstream, the Gulf Stream meets with the cold Newfoundland Current coming from the North, and splits into the North-Atlantic Subtropical Gyre and the North Atlantic Drift. In this region, marked in with the grey box in Figure 6, the bias changes sign compared to SEAS4.

5 This region is characterised by complex interactions of several large-scale ocean currents that are key to the North Atlantic ocean circulation (Buckley and Marshall, 2016). As will be seen in Section 5, SEAS5 also has reduced skill in this region compared to SEAS4.





To investigate the changes in this region, in Figure 7, we show a time-series of the mean SST in the region highlighted by the grey box in Figure 6 (50 to 30°W, 40 to 60°N). In ERA-Interim, the North Atlantic exhibits clear decadal variability: generally cold anomalies in the 1980's, warm anomalies in the 1990's and 2000's, and cold anomalies after 2010. SEAS4 captures this variability, showing the transition from cold to warm anomalies in the mid-1990's, while SEAS5 does not show

this variability, leading to a much lower anomaly correlation with respect to ERA-Interim in SEAS5 (-0.1) than in SEAS4 (0.8). Initial investigations suggest that this degradation is caused by the new ocean initial conditions (ORAS5), and is related to the increased horizontal resolution of the ocean analysis system. The deterioration of skill in this region can potentially affect forecasts over Europe through advection by the prevailing westerly winds. Studies further investigating the source and impact of this error are underway, and their results will be discussed in future publications.

To analyse changes in the extratropical atmosphere mean state, we first examine the zonally averaged temperature and wind profiles. Figure 8 shows the bias with respect to ERA-Interim in SEAS4 (c,d) and SEAS5 (e,f) for both DJF and JJA. The temperature profile is shown in the colours, while the zonal wind profile bias is over-plotted as contours. Improvements in model physics (see Section 2.2) have warmed the troposphere in SEAS5, which translates into a clear decrease in the bias in DJF, but in JJA the SEAS5 troposphere is too warm. The tropospheric warming from approximately 30°N to 40°N degrades

the JJA temperature gradients in SEAS5, and coincides with increased errors in the sub-tropical jets. The SEAS5 jets are too strong at the tropopause level in both seasons, but in JJA errors extend to lower levels and the jets are also positioned too far to the north in both hemispheres. Cold biases at the tropopause worsen in SEAS5, due in part to the increase in horizontal resolution in SEAS5 and in part to humidity errors at the tropopause (Polichtchouk et al., 2017; Shepherd et al., 2018).

To examine the spatial structure of these biases, in Figure 9 we show a map of 500 hPa geopotential height biases relative to

ERA-Interim in the northern extratropics. The warming of the troposphere in SEAS5 is reflected in higher geopotential heights in SEAS5, and in winter this substantially reduces the bias. In summer, the displacement of the jet shown in Figure 8 is clearly visible in SEAS5, but also present to a lesser extent in SEAS4.

Many of these bias patterns continue to the surface, as shown in a map of MSLP biases in Figures 10. In summer, SEAS5 high MSLP biases correspond to 500 hPa geopotential height biases in the northern Pacific and Atlantic. In winter, SEAS5

shows particular improvement (approximately 3 hPa) in the North Pacific subtropical high. There are also improvements in the winter MSLP trough that centred over the British Isles in SEAS4, however this is replaced with a bias that projects onto a negative North Atlantic Oscillation (NAO) pattern, reducing the gradient between the NAO centres of action. This may affect whether NAO events have the correct impact.

While the extratropics are less predictable on seasonal timescales than the tropics, it is common to analyse the performance

of a seasonal forecast system in predicting circulation patterns such as the North Atlantic Oscillation (NAO) and the Pacific-North America (PNA). In Figure 11, we show a time-series of a DJF NAO index using 25 ensemble members from SEAS4 and SEAS5, calculated by projecting DJF 500 hPa geopotential height onto the first EOF of ERA-Interim 500 hPa geopotential height in the North Atlantic (80°W to 40°E, 30° to 88.5°N) (Wallace and Gutzler, 1981) [5]. We see little difference between SEAS4 and SEAS5; both show moderate skill with an anomaly correlation of 0.44 in SEAS5 and 0.45 in SEAS4. Average

---

[5]Please note this NAO definition is similar but not identical to that used for the operational charts on ECMWF's website



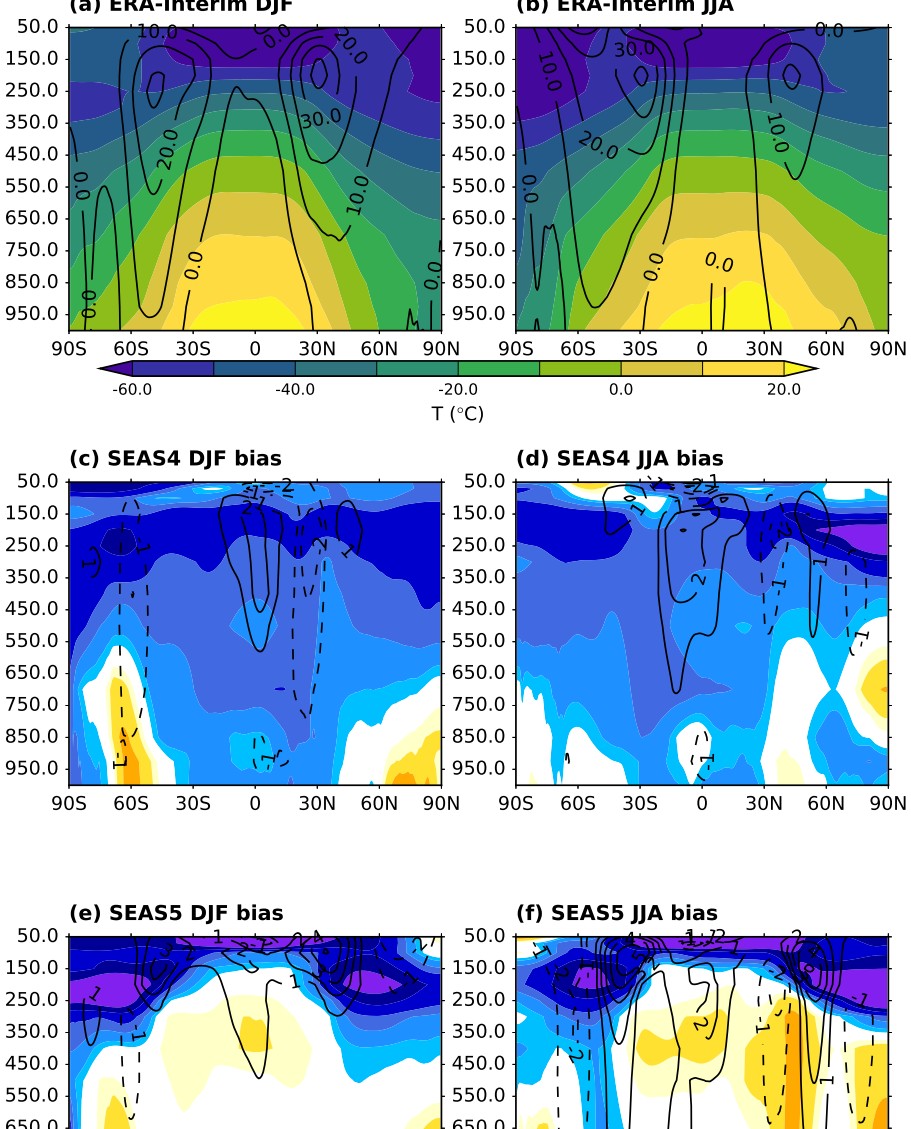

**Figure 8.** ERA-Interim zonally averaged profiles of temperature (colours) and zonal wind (contours) for DJF (a) and JJA (b), as well as the biases of SEAS4 (c,d) and SEAS5 (e,f) at one month forecast lead.

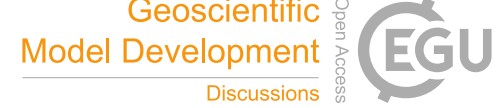



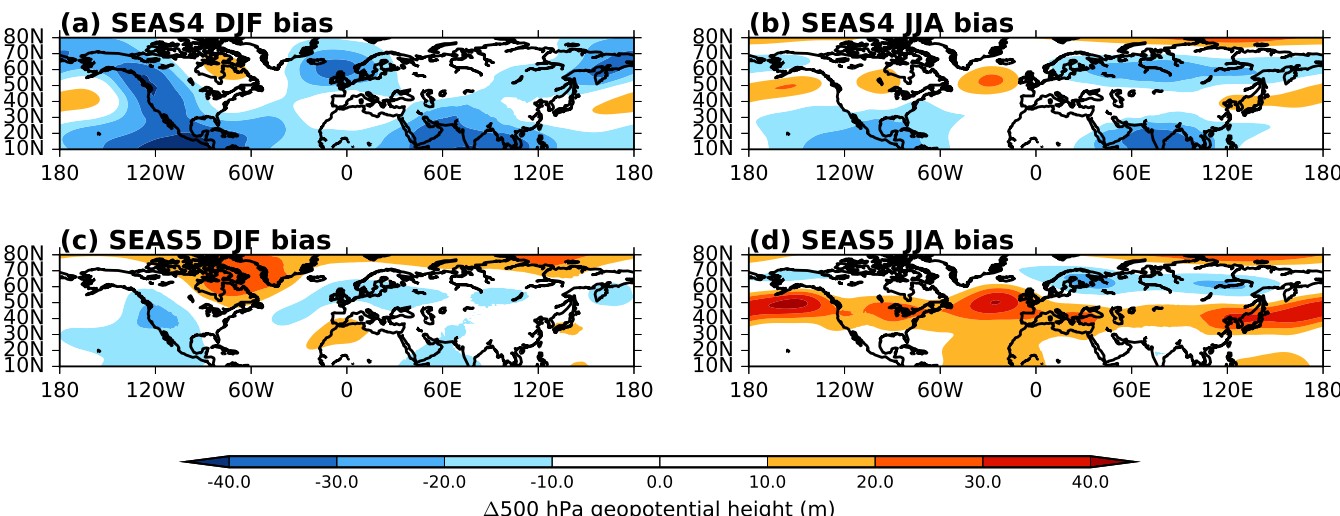

**Figure 9.** Northern extratropics winter and summer 500 hPa geopotential height bias with respect to ERA-Interim in SEAS4 (a,b) and SEAS5 (c,d) at one month forecast lead.

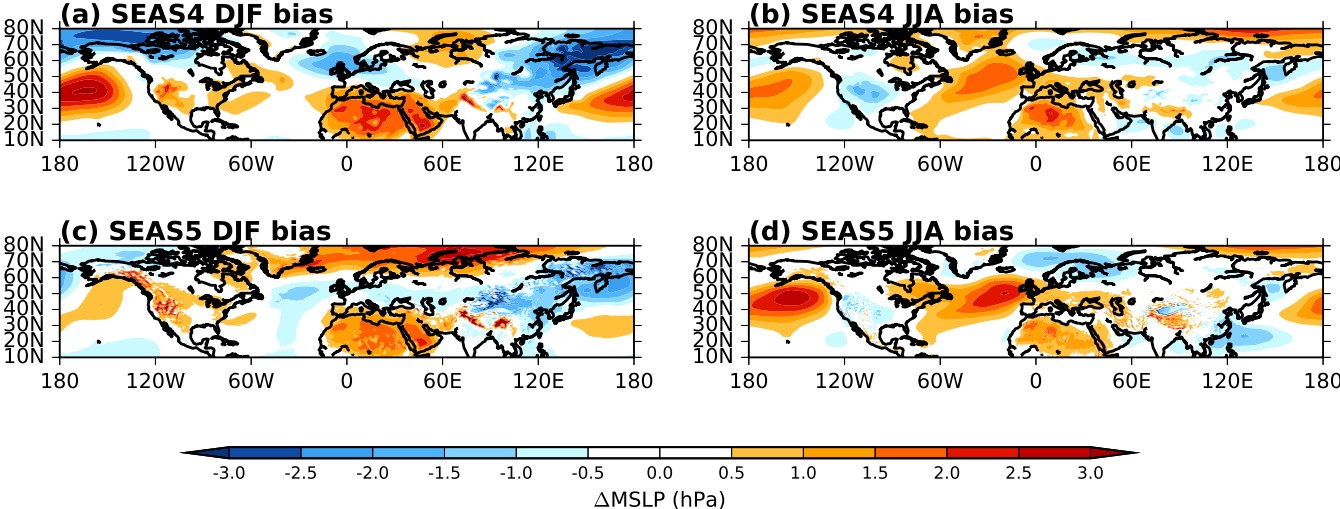

**Figure 10.** Northern extratropics winter and summer mean sea level pressure bias with respect to ERA-Interim in SEAS4 (a,b) and SEAS5 (c,d) at one month forecast lead.

ensemble spread (standard deviation) has similarly not changed between SEAS4 and SEAS5. Following Dunstone et al. (2016), we also calculated the NAO index as the MSLP difference between two small regions in the North Atlantic (Iceland: 25° to 20° W, 63° to 70° N and Azores: 28° to 20° W, 36° to 40° N) where we obtain an anomaly correlation of 0.30 in SEAS5 and 0.39 in SEAS4. This suggests the NAO in the ECMWF model may be less well represented at the surface and is also a reminder that statistics of the NAO are sensitive to which diagnostic is used, how it is calculated and which months and years are used in the calculation. For example, the confidence interval for sampling error over years is 0.12 to 0.67 for SEAS5. The errors in decadal

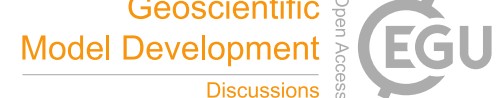



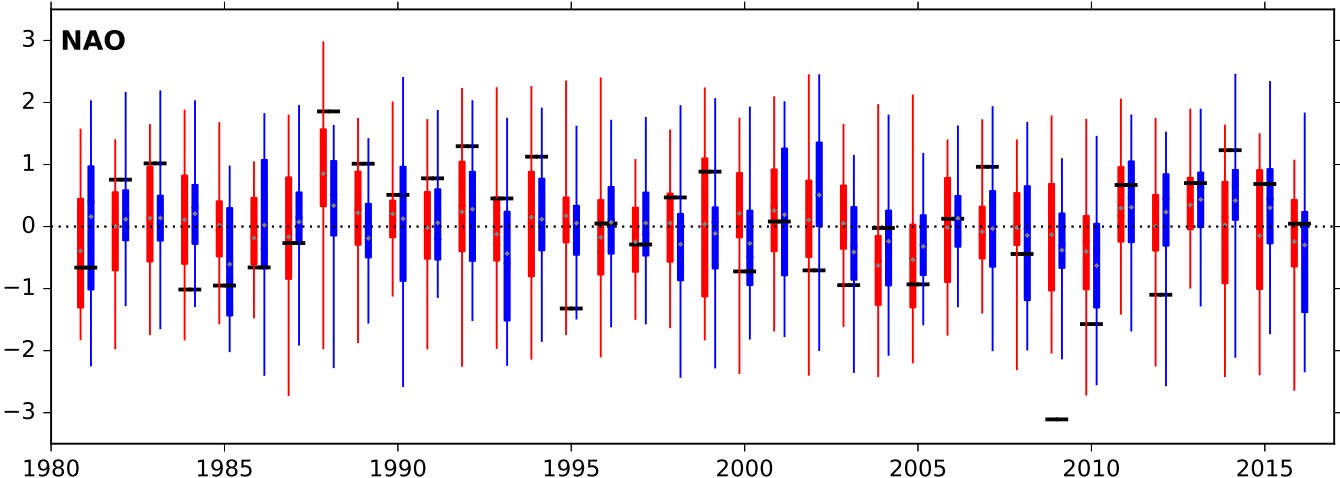

**Figure 11.** Time series of a DJF NAO index derived from projecting the re-forecast 500 hPa geopotential height onto the first EOF of ERA-Interim 500 geopotential height in the North Atlantic. Quartiles, minimum and maximum of the SEAS4 25 member ensemble are shown in blue, while the SEAS5 25 member ensemble is shown in red and ERA-Interim reanalysis is shown in the black bars. Forecasts were initialised in November, and the year shown is the year the ensemble was initialised. The grey diamonds indicate the ensemble mean. Anomaly correlation values for the ensemble mean are 0.45 for SEAS4 and 0.44 for SEAS5. The 95% confidence interval for sampling error over years is 0.12 to 0.67 for SEAS5.

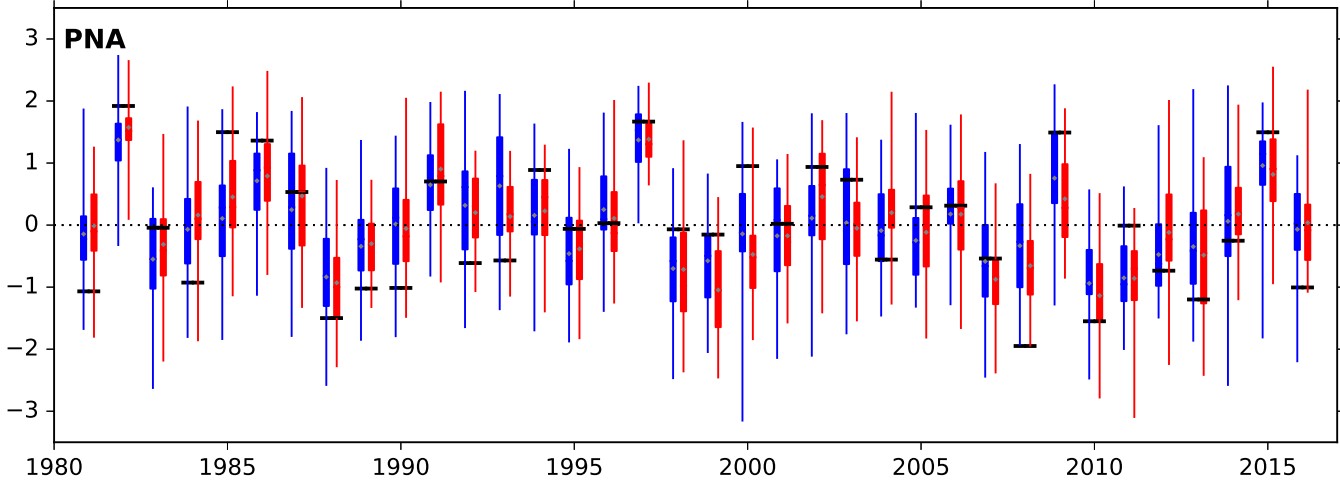

**Figure 12.** As in Figure 11, but for a PNA index. Anomaly correlation values for the ensemble mean are 0.69 for both systems. The 95% confidence interval for sampling error over years for SEAS5 is 0.47 to 0.83.

variability in the Northwest Atlantic discussed earlier may have a downstream impact on NAO skill in SEAS5, investigations are ongoing.

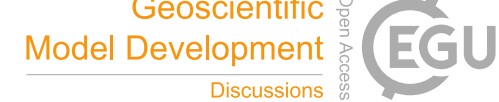



We show a Pacific-North America (PNA) teleconnection index in Figure 12, derived by projecting the DJF 500 hPa geopotential height onto first EOF of ERA-Interim geopotential height in a North Pacific/North American region (140°E to 80°W, 30° to 88.5°N). The skill predicting the PNA is much higher than the NAO, 0.69, but similarly there is little difference between SEAS4 and SEAS5, despite the improvements in ENSO prediction and the north Pacific MSLP bias in SEAS5. The correlation values for the PNA are less uncertain, for example, the confidence interval for sampling error over years is 0.47 to 0.83 for SEAS5.

Teleconnections from the tropics are an important source of predictable signals for the extratropical regions, and poor representation of teleconnections could be an explanation for low prediction skill in the extratropics. Although they can be detected throughout the whole yearly cycle, many teleconnection patterns affecting the northern mid-latitudes reach their largest amplitude during the boreal winter, when the strong vorticity gradients in the subtropical regions intensify the Rossby wave sources associated with tropical convection (Sardeshmukh and Hoskins, 1988).

A detailed analysis of SEAS4 teleconnections originating from tropical Indo-Pacific rainfall anomalies during the northern winter was carried out by Molteni et al. (2015, MSV15 hereafter). Overall, SEAS4 provided a good simulation of the relationship between SST and rainfall anomalies within the tropical belt, and of extratropical teleconnections to the North Pacific – North American sectors. On the other hand, teleconnections to the Euro - Atlantic sector in SEAS4 showed significant differences from the corresponding observed patterns, with an underestimation of the link between western/central Indian Ocean rainfall and NAO variability, and an incorrect phase of the ENSO response over the North Atlantic (see Fig. 6 in MSV15). The latter problem was linked to an excessively strong correlation between rainfall anomalies around the Niño 4 region (160°E to 150°W, 10°S and 10°N, Niño 4w) and the western/central Indian Ocean (40 to 90°E, 10°N to 10°S, WCIO).

Although a more detailed analysis of teleconnections in SEAS5 will be provided in other publications, here we summarise preliminary results:

- Connections between tropical SST and tropical rainfall show relatively minor changes with respect to SEAS4; this implies an overall satisfactory SEAS5 performance, but also the persistence of the too strong correlation between Niño 4w and WCIO rainfall (see Figure 13).

- Teleconnections into the Euro-Atlantic sector show bigger differences from SEAS4, with an improved pattern associated with central Pacific precipitation anomalies, but a substantial failure in reproducing the NAO connection with WCIO rainfall (see Figure 14, to be compared with Fig. 6 in MSV15)

The reasons for both the improvements and deterioration of extratropical teleconnections in SEAS5 are still being investigated. The improved simulation of the ENSO response is consistent with the general improvements in the representation of ENSO reported in previous sections of this paper. The deterioration of the WCIO-North Atlantic connection is also evident in multi-decadal coupled simulations run for the PRIMAVERA project (Roberts et al., 2018) and performed with the same IFS and NEMO versions used in SEAS5. Simulations analogous both to SEAS5 and to the multi-decadal simulations which use prescribed, observed SST show a much better agreement with observations (Molteni, Roberts and Senan, private communication). Since links between Indian Ocean rainfall and the NAO are also evident on the sub-seasonal time scale (Cassou, 2008;





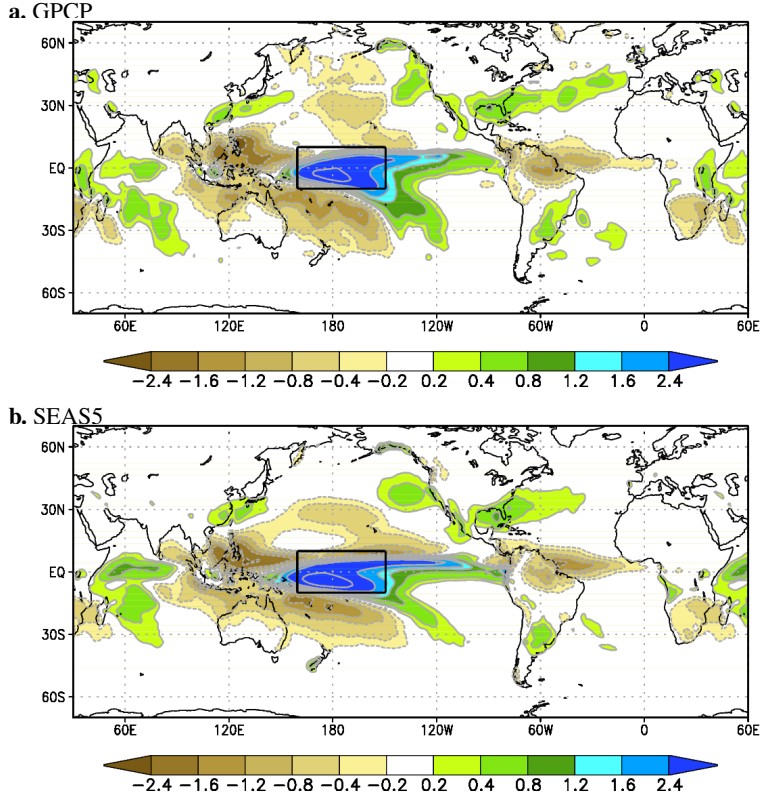

**Figure 13.** Covariance between normalised DJF rainfall anomaly in the Niño 4w region (black box) and rainfall anomaly elsewhere, following Molteni et al. (2015). Top: GPCP v2.3 data; bottom: SEAS5 re-forecasts. Note the stronger signal over the western Indian Ocean in SEAS5.

Lin et al., 2009), future analysis of SEAS5 performance in reproducing tropical intra-seasonal variability (such as the Madden-Julian Oscillation) and the associated ocean-atmosphere feedbacks may shed light on the causes of deficiencies detected on the seasonal scale.

### 4.3 Arctic

5  SEAS5 is the first seasonal forecast system at ECMWF to contain an interactive sea ice model. SEAS4 prescribed sea ice in re-forecasts and forecasts, using a scheme that sampled ERA-Interim data from the five previous years. Consequently, SEAS4 was able to capture the long term trends in sea ice evolution, but not the interannual variability of sea ice. Sea ice forecasts are relevant for industries such as shipping and fishing. Sea ice also has locally strong impacts on the forecasts of near surface parameters and may affect mid-latitude weather through teleconnections. The LIM2 model enables SEAS5 to forecast

10  interannual variability in sea ice concentration. However, the introduction of a fully prognostic sea-ice model introduces biases in the hindcast set. Seasonal Arctic sea ice biases for SEAS5 are shown in Figure 15, relative to the OSI SAF global sea ice





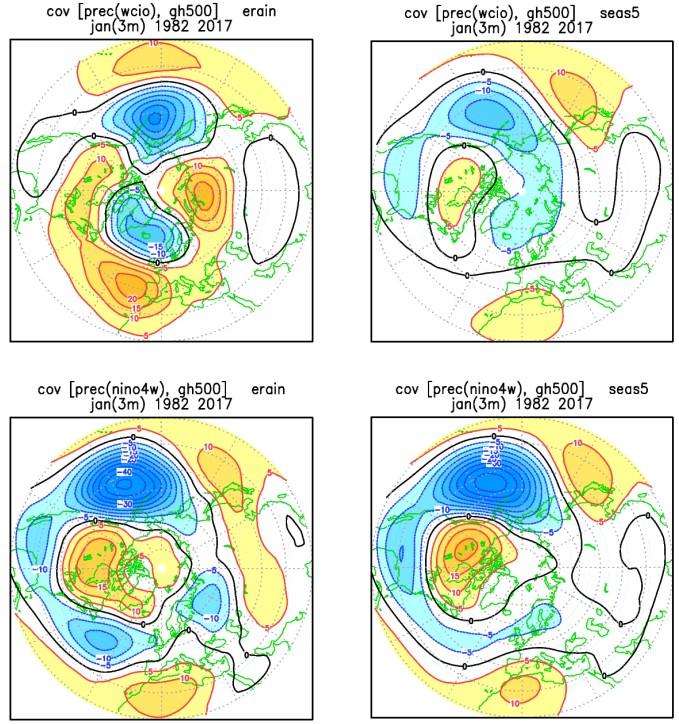

**Figure 14.** Covariances between normalised DJF rainfall anomalies in the western/central Indian Ocean (WCIO, top) and Niño 4w (bottom) regions, and 500-hPa height anomalies over the northern extratropics. Left column: from GPCP v2.3 rainfall and ERA-Interim geopotential height data; right column: from SEAS5 re-forecasts. SEAS4 results are shown in Fig. 6 in MVF15.

concentration climate data record (OSI-450, see Section 3.2 for details). The most noticeable biases are excess sea ice in the summer, due to not enough seasonal melting of the ice in SEAS5, and a lack of sea ice in the autumn, due to slow re-freezing of the ice. In spring, summer and winter there is excess ice in the Greenland Sea, along the Odden ice tongue. This bias is caused by ice that remains in later decades in SEAS5, which is rarely present after the 1990's in reanalysis.

5    Despite introducing these biases, including the interactive sea ice model improves the skill in predicting the interannual variability of sea ice. This is illustrated in the sea ice concentration RMSE maps shown in Figure 16. As with other variables, sea ice concentration is bias corrected before calculation of RMSE, but as sea ice concentration is a value that varies between zero and one, grid points were not bias corrected to values greater than one or less than zero. The RMSE in SEAS5 is typically 10-25%, which is an improvement over SEAS4 of 1-3% in many locations and up to 5% in some places. The largest improvements

10   are seen in autumn, probably because autumn is the season most affected by interannual variability. There are regions where the RMSE increases, such as in Bering Straight and Okhotshk Sea in summer and in the location of the Odden ice tongue bias in spring, but overall LIM2 is having a positive effect on forecasts of sea ice anomalies.





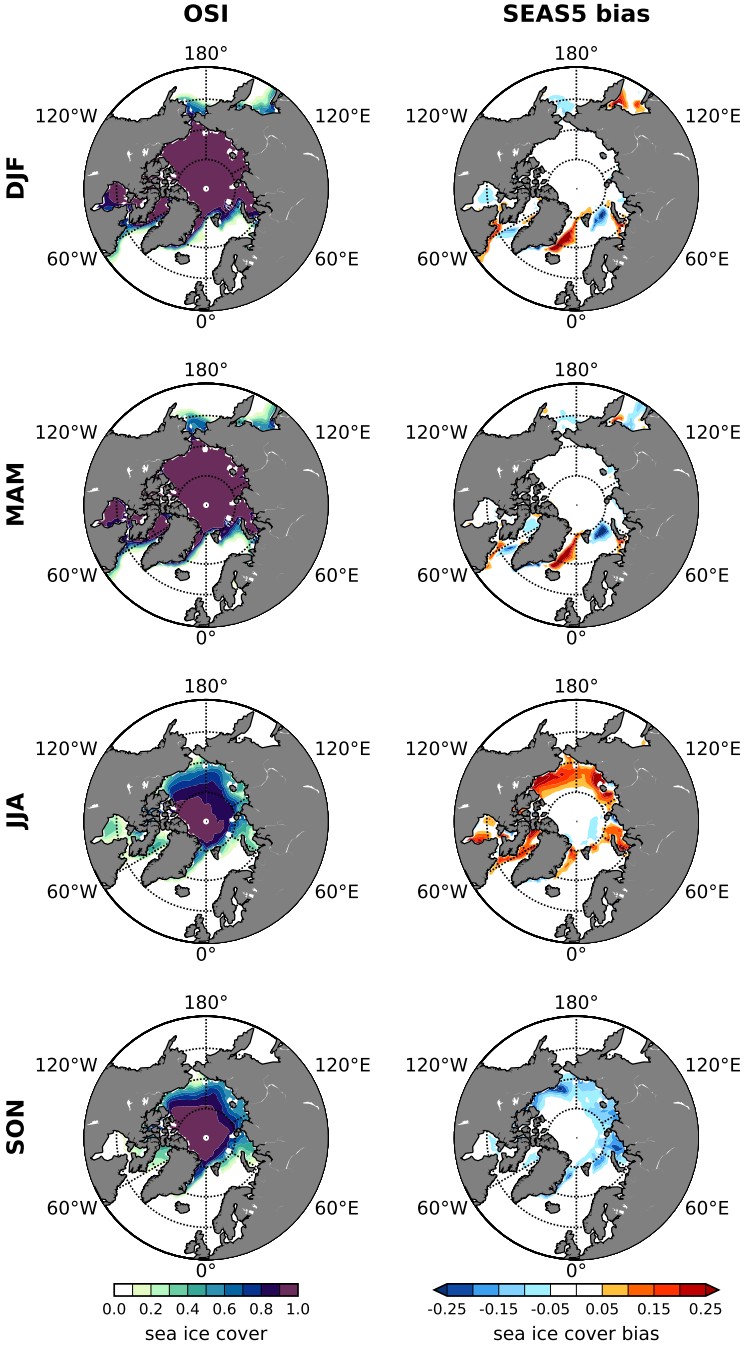

**Figure 15.** SEAS5 arctic sea ice concentration biases (right), relative to OSI-450 climatology (left) at one month lead time. Due to gaps in the satellite record, a small number of seasons had to be excluded from this analysis, see Section 3.2 for details.



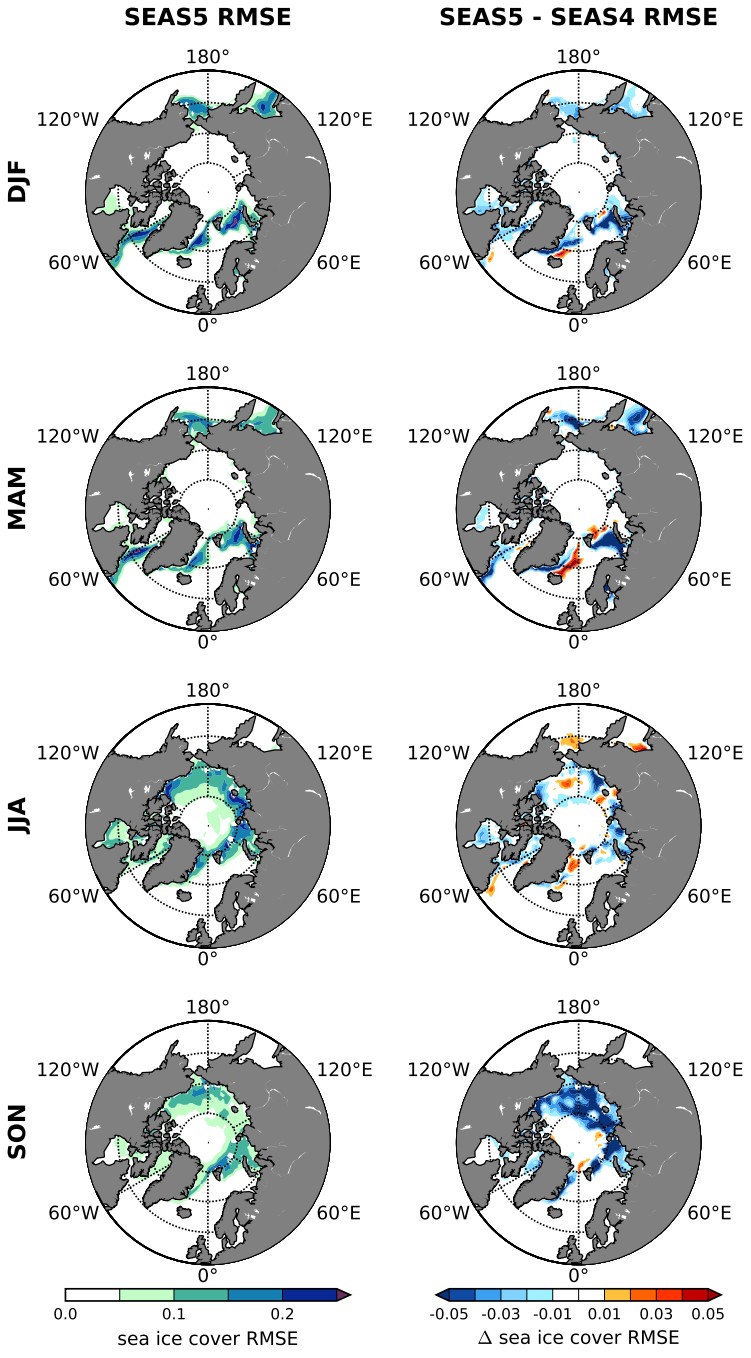

**Figure 16.** SEAS5 bias-corrected seasonal arctic sea ice concentration RMSE maps (left) relative to OSI-450, and the difference compared to SEAS4 (right) at one month lead time. Twenty-five ensemble members are used from each forecast system. Due to gaps in the satellite record, a small number of seasons had to be excluded from this analysis, see Section 3.2 for details.

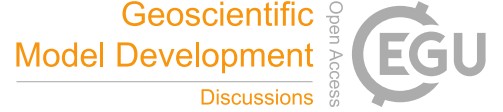

## 4.4 Stratosphere and QBO

Dynamical processes in the stratosphere are increasingly seen as a possible source of seasonal predictability. Teleconnections from the tropical oceans to the mid-latitudes may be mediated by the stratosphere (Bell et al., 2009; Ineson and Scaife, 2009), placing importance on correctly representing the mean stratosphere climate (Maycock et al., 2011). Additionally, the quasi-

biennial oscillation of the tropical stratosphere (QBO, Reed et al., 1961) potentially provides one of the few purely atmospheric sources of predictability on the seasonal timescale (e.g. Ebdon, 1975; Folland et al., 2012).

Figure 17 shows DJF zonal wind and temperature profiles in the troposphere and stratosphere for SEAS4 and SEAS5, extending the profiles shown in Figure 8 to 1 hPa. As discussed in Section 4.2, SEAS4 had a pervasive 0.5°C to 2.0°C cold bias to a height of about 20 hPa, with a warm bias above it. The cold bias has disappeared in the troposphere in SEAS5, but

increased to as much as 5°C in the lower stratosphere, just above the tropopause. At about 10 hPa in the tropics, this cold bias transitions to a warm bias in the upper stratosphere. These changes correspond to a steepening of the temperature gradient from the lower to upper stratosphere. The tropopause cold temperature bias in SEAS5 is accompanied by errors in the mid-latitude jets at the tropopause level (see Section 4.2) and excess equatorial westerly wind biases are present above 40 hPa. These excess winds were also present in SEAS4, but have worsened in SEAS5. The boreal winter-time polar vortex weakens in SEAS5,

resulting in an easterly bias throughout the depth of the stratosphere. Corresponding bias changes are seen in JJA: both in the winter (southern) hemisphere and the tropics.

As mentioned in Section 2.2, the tropical non-orographic gravity wave drag was reduced from its default value in IFS cycle 43r1 in order to improve the representation of the QBO in SEAS5. To illustrate the motivation and result of this change, in Figure 18 we compare the amplitude and phase of a QBO index as a function of lead time for SEAS4, the default cycle

43r1 IFS, and SEAS5 as the solid lines. We also show the annual range forecast for SEAS4 and SEAS5 only as the dashed lines. To compare SEAS4 and SEAS5 fairly with the smaller dataset available for the default IFS cycle 43r1, we use only five ensemble members with initialisation dates from 1993 to 2015, while the annual range forecasts contain 15 ensemble members with initialisation dates from 1981 to 2016. This illustrates that the number of ensemble members and years has some effect on statistics of QBO skill, as shown by the differences between the dashed and solid lines. We use the monthly zonal wind

averaged from 5°N to 5°S at 30 hPa as a QBO index.

The anomaly correlation of the default IFS cycle 43r1 decreases sharply after month two in forecasts initialised in both May and November. For the November initialisation the correlation is comparable to, or just exceeding, a persistence forecast (not shown), while for the May initialisation the first few months exceed persistence. In contrast, SEAS5 improves on SEAS4, and has an anomaly correlation exceeding 0.7 throughout the long range forecast. The QBO amplitude is lower in SEAS4 than in

reanalysis, and this reduces even more in SEAS5. Reducing the tropical non-orographic gravity wave drag does not improve the amplitude of the QBO; in the forecast initialised in November it even degrades the amplitude further. However, the combination of the improvement in anomaly correlation and the degradation in amplitude results in a comparable QBO RMSE in SEAS4 and SEAS5, whereas the default IFS cycle 43r1 has a much larger RMSE. Reducing the tropical non-orographic gravity wave drag also reduces the equatorial zonal wind bias around 10 hPa (not shown), though a large bias remains. However, lower in



**Figure 17.** Zonally averaged profiles of ERA-Interim DJF zonal temperature (a) and wind (b), as well as biases in SEAS4 (c,d) and SEAS5 (e,f) at one month forecast lead.





**Figure 18.** Metrics summarising the phase and amplitude of a QBO index at 30 hPa in SEAS4 (blue), SEAS5 (red) and IFS cycle 43r1 with default settings (green) relative to ERA-Interim reanalysis for forecasts initialised in May and November. Top panels: Anomaly correlation, middle panels: ratio of the standard deviation of the system to the standard deviation of ERA-Interim reanalysis, bottom panels: RMSE. To compare SEAS4 and SEAS5 fairly with the data available for the default 43r1 cycle, the solid lines use only five ensemble members with initialisation dates from 1993 to 2015. The dashed lines compare SEAS4 and SEAS5 for the entire 13 month duration of the annual range forecasts from 1981 to 2016, using the 15 ensemble members.

the stratosphere the QBO deteriorates in both phase and amplitude compared to SEAS4 (not shown), despite the reduction in the tropical non-orographic gravity wave drag.

As lead time increases, the QBO amplitude in SEAS5 decreases and worsens relative to SEAS4. The RMSE in the annual range forecast is large in both systems, though it shows more skill than a persistence forecast. Differences between the SEAS4



and SEAS5 anomaly correlations of the annual range forecast depend on season and lead time. Differences in RMSE in the annual range forecasts are small compared to the RMSE.

Although predicting the QBO phase is potentially important for improving the seasonal forecast skill, the realisation of this skill relies on the teleconnections between the QBO and the extratropics which are generally not very well represented in seasonal systems (Scaife et al., 2014). Future work will evaluate this teleconnection in SEAS5.

## 5   SEAS5 verification: Skill and reliability of user-relevant parameters

In the previous section we discussed the performance of SEAS5 from the perspective of model development and predictability. In this section, we present verification metrics corresponding to the SEAS5 operational charts and focus on the skill of variables more relevant for users: two-metre temperature and precipitation. Here, we discuss JJA and DJF at one month lead time. A more comprehensive set of seasonal forecast skill measures for all seasons, lead times and additional atmospheric variables is available on ECMWF's website.[6]

We first use maps of the temporal anomaly correlation of the forecast ensemble mean anomalies with the observed anomalies to show the geographical distribution of skill over the globe. The use of deterministic skill measures such as the anomaly correlation of the ensemble mean, is common practise despite the probabilistic nature of the seasonal predictions. To add a probabilistic measure of skill we then discuss differences in continuous ranked probability skill score (CRPSS) between SEAS4 and SEAS5. Finally, we discuss SEAS5 two-metre temperature reliability over the tropics and Europe.

As discussed in Section 3, the re-forecast set we evaluate here has 25 members, while operational forecasts have 51 members. Consequently, the skill estimates based on the re-forecasts are a systematic underestimate of the expected skill of the operational ensemble, although a real-time system also carries a slightly higher risk of issues such as unexpected changes in observing systems or unpredicted changes in climate system behaviour.

### 5.1   Anomaly correlation

In Figure 19 the geographical distribution of two-metre temperature skill at one month lead time is represented by the local correlation between ensemble-mean of the re-forecasts and ERA-interim. High skill for near-surface temperature is evident over the tropics, particularly over the tropical oceans where skill reaches a maximum in the central and east Pacific. A number of extra-tropical regions, depending on the season, also show a useful of skill. In winter, SEAS5 shows some level of skill across northern and central Europe, with areas of significance over Scandinavia. In summer, we see significant skill over southeastern Europe and the Mediterranean. Some of this skill is associated with the model's ability to represent the longer term trends (decadal variability and climate change) as well as its ability to correctly forecast interannual variability.

Forecast skill is low in places over continental North America and Eurasia, which is common in seasonal forecast systems (Molteni et al., 2011; Kim et al., 2012; Maclachlan et al., 2015), and is also evident in MSLP and 500 hPa geopotential skill (not shown). There is a region over the northwest Atlantic by the Grand Banks of Newfoundland with negative correlation

---

[6]More charts available at: https://www.ecmwf.int/en/forecasts/charts/catalogue/?facets=Range,Long%20(Months)





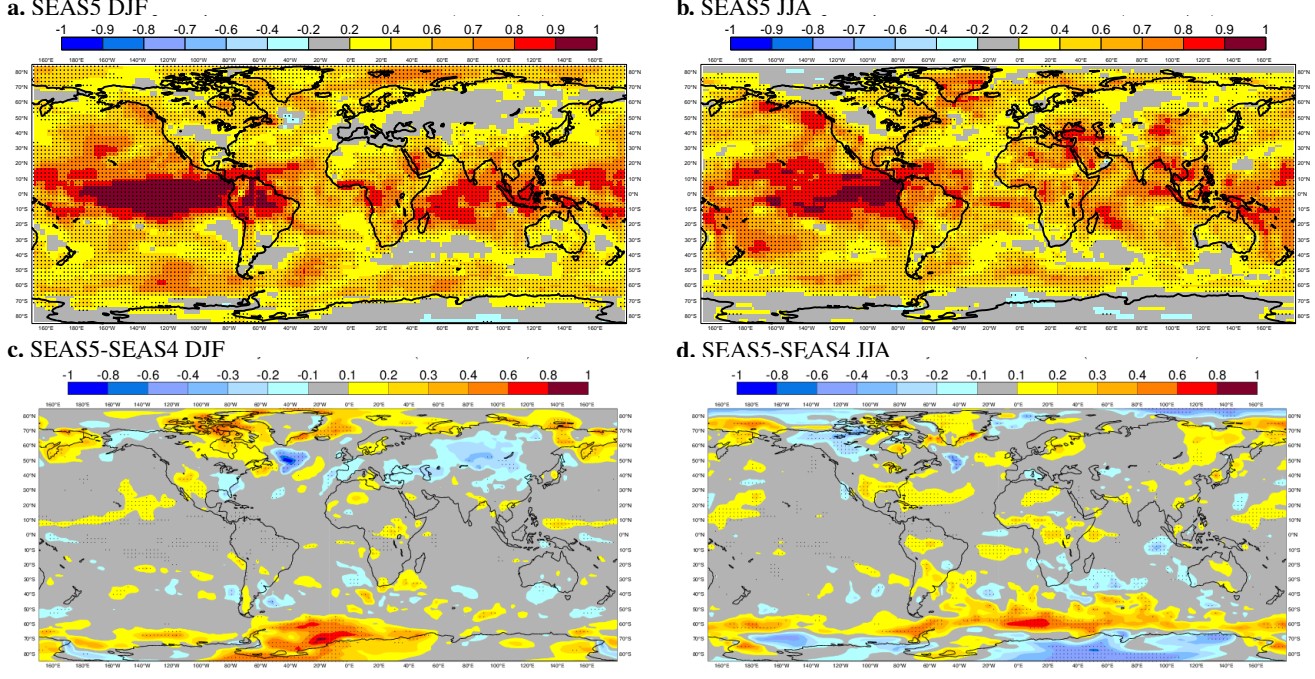

**Figure 19.** Top: Anomaly correlation map of the ensemble-mean SEAS5 mean two-metre temperature forecast for DJF (left) and JJA (right) at one month forecast lead. Two-metre temperature is verified with ERA-Interim data. Locations with correlation values different from zero at the 5% significance level are highlighted by dots. Bottom: Difference between SEAS5 and SEAS4 anomaly correlation, with 25 ensemble members used in each. Locations where the correlation values are different at a 5% significance level are highlighted by dots.

values in winter. As discussed in Section 4.2, SEAS5 poorly captures the observed decadal variability of the north Atlantic subpolar gyre, which decreases skill in this region. There are also skill minima over other ocean boundary currents, though little is known about the potential predictability in these regions.

Figure 19 also shows the difference in two-metre temperature skill between SEAS5 and SEAS4. Improvements in winter
5   are found in the tropical and sub-tropical Eastern Pacific reaching the west coast of America, likely associated with the improvements in ENSO bias and variability discussed in Section 4.1. A degradation is seen in the EEIO in JJA, due to the errors in EEIO variability also discussed in Section 4.1. In JJA, significant improvements in skill are seen over equatorial Africa and equatorial North and South America.

In the extratropics, some improvement in summer skill is found over Greenland and eastern Siberia. Figure 19 also shows
10  the decrease in skill over the northwest Atlantic (Section 4.2). There is no evidence of locally significant skill improvement over Europe in either season. There is an overall improvement in two-metre temperature predictions north of 60°N and south of 60°S in both seasons, including a substantial enhancement of JJA skill is found in the southern hemisphere. This is likely





related to the improved predication of sea ice concentration generated by the addition of LIM2. At longer time ranges (month 5 to 7, not shown) SEAS5 exhibits enhanced skill over the tropical oceans (0° to 20°N).

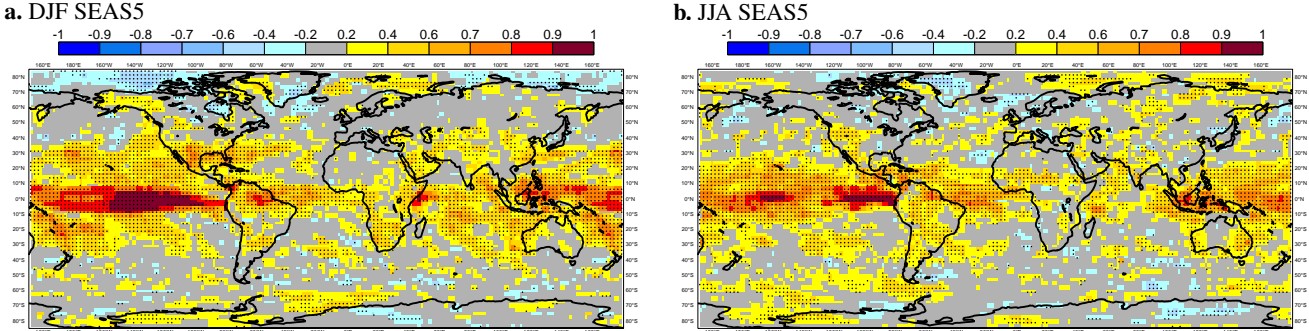

**Figure 20.** Anomaly correlation maps of the SEAS5 ensemble-mean precipitation forecast for DJF (left) and JJA (right) at one month forecast lead. Precipitation is verified with GPCP v2.2 data from 1981 to 2014. Locations with correlation values different from zero at the 5% significance level are highlighted by dots.

In Figure 20, we show the anomaly correlation maps of SEAS5 precipitation compared to GPCP2.2 (1981 to 2014). Precipitation skill is much noisier and significantly lower than the skill for near-surface temperature. Over the tropical oceans the signal looks more coherent, while seasonal prediction for rainfall over land generally has lower skill, even in the tropics. "Local" (i.e. grid-point) seasonal rainfall predictions often have limited skill, but spatially averaged values over many tropical regions have significant predictability and play a crucial role for extratropical predictability (Molteni et al., 2015; Scaife et al., 2017, 2018). Differences in rainfall anomaly correlation between SEAS4 and SEAS5 are noisy, but more coherent patterns are present in a few locations (not shown). In summer, the precipitation skill is improved substantially over New Guinea relative to SEAS4. This improvement continues into the later months of the forecast (forecast months 5 to 7), to show a large increase in skill over New Guinea and the tropical west Pacific in the autumn. A more detailed analysis of precipitation skill differences would be useful in future, but for the remainder of this section we focus on two-metre temperature, where differences between SEAS4 and SEAS5 are more distinct.

## 5.2 CRPSS

It is common to use CRPSS, the skill score version of the continuous ranked probability score (CRPS, Hersbach, 2000; Matheson and Winkler, 1976; Wilks, 2011) to evaluate the benefit of a forecasting system. The CRPS is the integral of the Brier score over all possible threshold values, for a given variable. For a deterministic forecast the CRPS reduces to the mean absolute error. The CRPSS then gives an indication of the added value of a forecasting system over simply forecasting climatology, a value of one indicating perfect forecasts, zero showing no improvement over climatology and negative values indicating a failing forecasting system. In Figure 21, we use maps of two-metre temperature SEAS5 CRPSS relative to ERA-Interim and

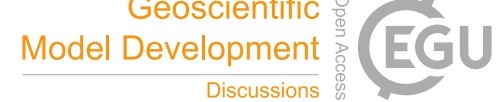



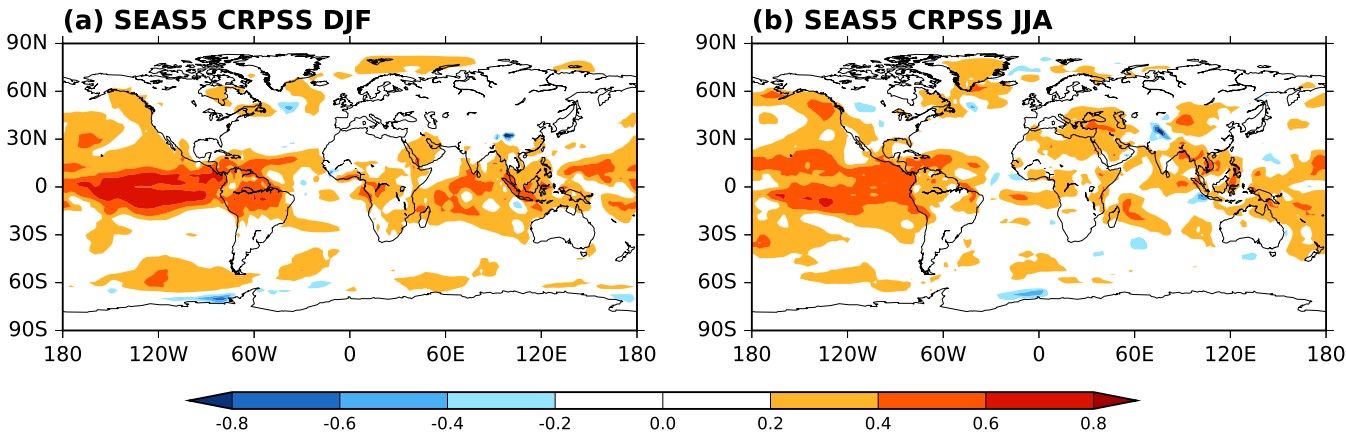

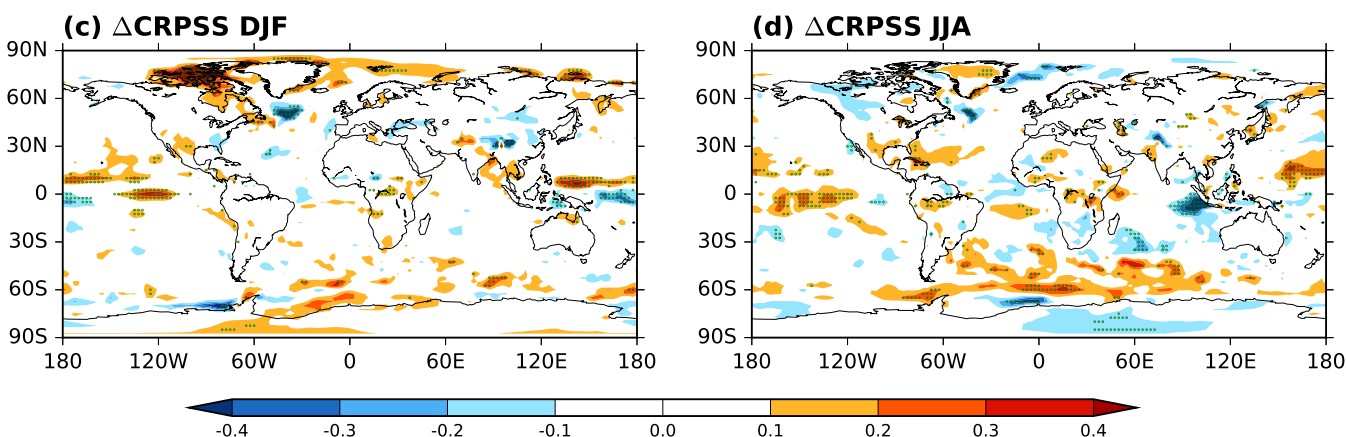

**Figure 21.** Top: CRPSS maps of two metre temperature skill for SEAS5 with reference to ERA-Interim climatology in (a) DJF and (b) JJA. Bottom: The change in two-metre temperature CRPSS score between SEAS4 and SEAS5 in (c) DJF and (d) JJA. Green stippling is plotted at $p < 0.05$.

CRPSS differences between SEAS4 and SEAS5 to highlight the changes in probabilistic skill between SEAS5 and SEAS4 (see Section 3.1 for a description of how the CRPSS is calculated).

SEAS5 CRPSS relative to ERA-Interim demonstrates that in the tropics, SEAS5 generally provides improved skill compared to ERA-Interim climatology. Outside the tropics, there are a few, seasonally dependent, regions where SEAS5 skill exceeds
5   climatology. Regions with with negative scores are small, but often correspond to known deficiencies in the system including in the North Atlantic in winter and the EEIO in summer.

The changes in CRPSS broadly agree with the changes in anomaly correlation seen in Figure 19. The tropical Pacific shows improvement across the basin in JJA. In DJF, this improvement extends from 120°E to 120°W, but is confined to the North of the equator, while to the south there is some deterioration. The improvement seen in eastern equatorial Africa in the

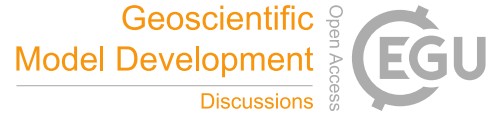



**a.** Tropics
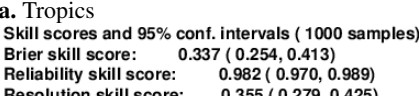

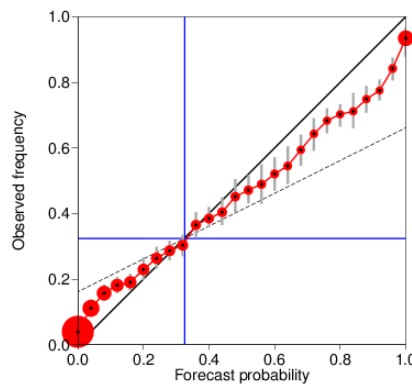

**b.** Europe (land and sea)
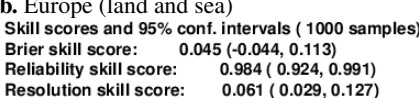

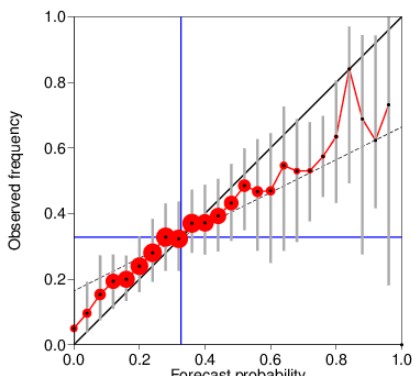

**Figure 22.** DJF Two-metre temperature reliability diagrams for SEAS5, computed including all grid points for the tropics (left, 20°N to 20°S) and a European region (right, 35° to 75°N, 12.5°W to 42.5°E) over the whole re-forecast period using 25 ensemble members. Verification data is ERA-Interim. Reliability diagrams are computed for three month average forecast anomalies in the upper third of the model climate distribution. Grey vertical lines indicate the 95% confidence intervals.

anomaly correlation maps is also present in CRPSS. The deterioration in the North Atlantic and EEIO is also evident. Over the Himalayas little decrease in skill is seen in the anomaly correlation maps, but a clear decrease is seen in CRPSS. This may indicate a change in the spread in SEAS5, but further analysis is needed to understand this feature. Areas of significant improvement and deterioration are evident in the northern and southern hemispheres around the edges of the sea ice, though
5    not as widespread as in anomaly correlation.

## 5.3 Reliability

Reliability measures the ability of a forecast system to represent the observed frequency of events. Reliability is an important consideration for the usefulness of probabilistic predictions, as a user might be able to make use of a forecasting system with limited skill if the system is statistically reliable. Reliability is typically illustrated using reliability diagrams (see Section 3.1.5).
10   In a perfectly reliable system, the forecast probability will equal the frequency of occurrence and points will lie along straight diagonal line. Figure 22 shows reliability diagrams for warm two-metre temperature anomalies in DJF. Forecasts in the tropics (20°N to 20°S) show a small but systematic discrepancy between the forecast probabilities and observed frequencies where forecast probabilities for the event are further from climatology than is observed. This is a common property of seasonal forecast systems, often referred to as 'overconfidence' (Weisheimer and Palmer, 2014). Over Europe and its surrounding seas
15   (35° to 75°N, 12.5°W to 42.5°E), the forecast also tends to be overconfident, though reliability over land points is lower than

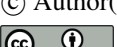


reliability over sea points in this region. Comparisons with SEAS4 reliability indicate only small changes in reliability between SEAS4 and SEAS5 (not shown).

## 6   Conclusions

ECMWF's fifth seasonal forecast system, SEAS5, became operational in November 2017, replacing its predecessor SEAS4.
SEAS5 features upgraded versions and increased resolution of the atmosphere and ocean models as well as adding the interactive sea ice model LIM2. It also represents a step towards a seamless system, with very few differences from the cycle 43r1 extended range (monthly) forecast system.

SEAS5 improves over SEAS4 in a number of ways. There is clear improvement in equatorial Pacific SST bias and in the cold bias present throughout the troposphere in SEAS4. SEAS5 skill in ENSO prediction increases, especially in the Western
- Central Pacific and in the annual range forecast, improving on already high skill in SEAS4. In spite of this noticeable skill improvement, ENSO forecast remain overconfident (underdispersive). The interactive sea ice model gives SEAS5 the ability to forecast sea ice concentration. This leads to improved predictions of Arctic sea-ice, and this improves two-metre temperature prediction skill around the sea ice edge.

Other aspects of SEAS5 are degraded compared to SEAS4. The variability in the eastern equatorial Indian ocean is very
overactive in SEAS5, posing a problem for teleconnections originating there. Skill has decreased in the northwest Atlantic where SEAS5 fails to capture multidecadal variability, an error which was not present in SEAS4. Temperature biases in the lower stratosphere and jets at the tropopause level are also degraded in SEAS5 relative to SEAS4, which could be inhibiting teleconnections and preventing increased tropical skill from generating increased extratropical skill. These issues are actively being investigated in order to improve future seasonal forecast systems.

Overall, SEAS5 is another step in the development of seasonal forecast systems at ECMWF, with advances generating higher levels of skill where expected (e.g. interactive sea ice), while some known deficiencies remain and others appear. SEAS5 continues to be a state-of-the-art seasonal forecast system, with a particular strength in ENSO prediction. SEAS5 data is contributed to the Copernicus climate change service's (C3S) multi-system seasonal forecast and is publicly available from the C3S climate data store along with data from other state of the art seasonal forecast systems. This creates opportunities for
a wide range of research on seasonal forecasting and predictability and could also be a catalyst for future seasonal forecast development.

*Code and data availability.*   The model configurations described here are based on the ECMWF Integrated Forecast System (IFS) and the NEMO/LIM ocean-sea ice model. The IFS source code is available subject to a license agreement with ECMWF. ECMWF member-state weather services and their approved partners will be granted access. The IFS code without modules for data assimilation is also available
for educational and academic purposes as part of the OpenIFS project (https://software.ecmwf.int/wiki/display/OIFS/OpenIFS+Home). The NEMO/LIM source code is available under a CeCILL free software license (https://www.nemo-ocean.eu/).





The reforecasts from SEAS5 are publicly available from ECMWF's Copernicus Climate Change Service (C3S), though its Climate Data Store (https://cds.climate.copernicus.eu/). Instructions on how to access this data are available from the C3S user support. SEAS4 is not a public dataset; to access SEAS4 data please contact the authors with a specific request.

*Competing interests.* The authors have no competing interests.

5 *Acknowledgements.* We would like to acknowledge all ECMWF staff who have actively contributed to the development of the IFS model and the successful implementation of SEAS5. The authors specifically acknowledge Richard Forbes, who contributed a written description of the IFS model differences between cycle 36r4 and cycle 43r1 to this article.

This study has been partially funded by the Copernicus Climate Change Service. ECMWF implements this Service and the Copernicus Atmosphere Monitoring Service on behalf of the European Commission.



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
