# Peer review of "SEAS5: The new ECMWF seasonal forecast system"

_Geoscientific Model Development, 2018_

## Referee Comment (RC1) · Anonymous Referee #1 · 30 Oct 2018

The paper describes the SEAS5 seasonal forecast system and documents its performance compared to its predecessor. SEAS5 has a number of enhancements compared to SEAS4, such as substantial increases in resolution of the atmosphere and ocean models and the introduction of a prognostic sea-ice model. The authors provide a very balanced evaluation of biases and skill, noting both improvements and deficiencies in the forecast system and highlighting aspects requiring future research. The range of diagnostics employed provides a good overview of performance, aptly setting the scene for other, more focussed studies on specific elements of the prediction system.

The paper is a valuable resource for both users of SEAS5 and other model and seasonal forecast system developers wanting to understand, for example, how adding

complexity (such as increasing resolution or introducing a sea-ice model) might impact on the performance of seasonal predictions. Given that the outputs of ECMWF's seasonal prediction system are widely used, including through the Copernicus multisystem seasonal forecast service, the paper will be of interest to a wide range of climate scientists.

The manuscript was a pleasure to read. It is very well written, clear, concise and well-structured. The results are appropriately analysed and presented. I do not think that any major changes are required.

Minor comments:

There are numerous instances of usage of "summer" and "winter" when results relevant to both hemispheres are being discussed (e.g., abstract line 17; multiple occurrences in section 4.1, some instances in Section 5.1). Either preface with boreal/austral or use actual months (DJF/JJA/. . .).

Page 7, Lines 11-15: Are the forecasts also initialised with the seasonally varying ozone climatology?

Page 8, Section 2.4.1 and 2.4.2: How does the SEAS5 ensemble generation differ from SEAS4?

Page 8, Section 2.4.1: It is not clear to me how the 5-member ensemble analysis is related to or used to create the 25-member ensemble.

Page 11, Section 3.1.4: Define the abbreviation "CRPSS"

Figure 2: It would be good to add the ensemble spread to panel (d).

Page 16, line 13: Should read in a "positive" IOD event, a cold anomaly. . ..

Page 37: The configuration of SEAS5 appears to be very close to that of the ENS monthly system. Are there plans to run just one system, across the timescales?

[Figure]

I am curious – how much more computationally expensive is SEAS5 given the increases in complexity compared to SEAS4?

Technical corrections:

Page 2, line 28: Remove the double "all": "...to document all all of them...."

Page 6, line29: change "where as" to "whereas"

Page 11, line 8: change "If follows" to "It follows"

Page 16, line 2: "12 months" should be "11 months"

Page 20, line 23: change "Figures" to "Figure"

Page 32, line25: Remove the "a": "...also show a useful of skill."

Page 35, line 5: Remove the double "with": "Regions with with negative..."

Page 37, line 11: change "forecast" to "forecasts"

---

## Referee Comment (RC2) · Anonymous Referee #2 · 9 Nov 2018

General comments:

The manuscript describing ECMWF's new seasonal forecast system SEAS5 contains a comprehensive selection of evaluation metrics of model skill, biases, variability and teleconnections. It also presents in a fair manner the aspects in which the model performs better and worse than its predecessor SEAS4. This in my opinion is very useful information for the user. I think the paper is well structured, easy to follow and gives an adequate overview of the forecast system which will serve as a guiding document for many users, both scientists and non-scientists.

Specific comments:

Page 2 line 28: Remove one " all"

[Figure]

Page 5 line 17: Is there a reason for using CMIP5 and not the most recent CMIP6 forcings? Could this have an impact?

Page 7 line 10: Why is there a change from ERA-Interim to ECMWF operational analysis (is it superior?)

Page 8 lines 14-18: There are a few changes for the ocean/sea ice initialization during the hindcast period. Does this have an impact on skill/bias/variability?

Page 11 line 5: Is there a reason for not applying cross-validation for other indexes?

Page 20 lines 7-9 and figure 19c: There seems to be a bit of degradation in DJF temperature skill over the Iberian peninsula as compared to SEAS4. For particular regions availability of SEAS4 forecast data could be advantageous. Has SEAS4 operational forecast stopped or will it continue in the future?

Page 25 Fig 13. Is there a reason for using normalized precipitation anomaly in the black box instead of the more standard NINO indices?

Page 34 line 9 and Fig 20. Is there a reason for not showing precipitation skill difference between SEAS5 and SEAS4, they are even discussed. I would suggest including it for completeness.

---

## Author Comment (AC1) · 18 Dec 2018

We would like to thank the reviewers for their useful comments. We have attached a document containing our response to this comment. We will also submit another comment with a pdf showing the changes to the original article.

Thank you, Stephanie Johnson (for the authors)

Please also note the supplement to this comment: https://www.geosci-model-dev-discuss.net/gmd-2018-228/gmd-2018-228-AC1-supplement.pdf

**Supplement:**

Tuesday, 18 December 2018

Dear Sir/Madam,

We would like to thank the reviewers for taking the time to consider our manuscript and for their helpful feedback. Please find our replies to each review in red below.

Sincerely,

Stephanie Johnson (on behalf of all authors)

**Anonymous Referee #1**

The paper describes the SEAS5 seasonal forecast system and documents its performance compared to its predecessor. SEAS5 has a number of enhancements compared to SEAS4, such as substantial increases in resolution of the atmosphere and ocean models and the introduction of a prognostic sea-ice model. The authors provide a very balanced evaluation of biases and skill, noting both improvements and deficiencies in the forecast system and highlighting aspects requiring future research. The range of diagnostics employed provides a good overview of performance, aptly setting the scene for other, more focussed studies on specific elements of the prediction system.

The paper is a valuable resource for both users of SEAS5 and other model and seasonal forecast system developers wanting to understand, for example, how adding complexity (such as increasing resolution or introducing a sea-ice model) might impact on the performance of seasonal predictions. Given that the outputs of ECMWF's seasonal prediction system are widely used, including through the Copernicus multi-system seasonal forecast service, the paper will be of interest to a wide range of climate scientists.

The manuscript was a pleasure to read. It is very well written, clear, concise and well-structured. The results are appropriately analysed and presented. I do not think that any major changes are required.

Thank you for this nice feedback. Please see our responses to your comments in line below.

Minor comments:

There are numerous instances of usage of "summer" and "winter" when results relevant to both hemispheres are being discussed (e.g., abstract line 17; multiple occurrences in section 4.1, some instances in Section 5.1). Either preface with boreal/austral or use actual months (DJF/JJA/...).

Thank you for pointing this out, we have adjusted these sentences throughout the paper.

Page 7, Lines 11-15: Are the forecasts also initialised with the seasonally varying ozone climatology?

Yes, and we have clarified this in the article.

Page 8, Section 2.4.1 and 2.4.2: How does the SEAS5 ensemble generation differ from SEAS4?

We have added a few sentences in Section 2.4.1 and 2.4.2 summarizing the differences.

Singular vectors are applied to the initial conditions in both systems with settings corresponding to their model cycle. EDA perturbations were not applied in SEAS4, but are applied in SEAS5.

The stochastic model perturbations are generated by the same two schemes (SPPT and SKEB) in SEAS4 and SEAS5. The settings used in these schemes are adjusted in the model development process, but are consistent with their respective cycles.

There are several differences between SEAS4 and SEAS5 ocean initial condition perturbations, the main differences are in the perturbation repository and the introduction of two temporal decorrelation scales, for details see Zuo et al. 2017:

Zuo, H., Balmaseda, M. A., Mogensen, K., Tietsche, S., OCEAN5: the ECMWF Ocean Reanalysis System and its Real-Time analysis component, 2018, ECMWF Tech Memo 823.

Page 8, Section 2.4.1: It is not clear to me how the 5-member ensemble analysis is related to or used to create the 25-member ensemble.

Thank you for mentioning this, it was not clear in the text. Each forecast ensemble member is assigned an ocean reanalysis member to initialize from, by cycling through the ocean analysis members from member 0 to member 4. Then, additional SST perturbations, which are unique to each ensemble member, are added. We have added a few sentences to the text that we hope will clarify this.

Page 11, Section 3.1.4: Define the abbreviation "CRPSS"

Thank you for spotting this, we have corrected it.

Figure 2: It would be good to add the ensemble spread to panel (d).

We have added this as the dotted lines.

Page 16, line 13: Should read in a "positive" IOD event, a cold anomaly....

Thank you for spotting this, we have corrected it.

Page 37: The configuration of SEAS5 appears to be very close to that of the ENS monthly system. Are there plans to run just one system, across the timescales? I am curious – how much more computationally expensive is SEAS5 given the increases in complexity compared to SEAS4?

There are complexities that would need to be addressed to run one system across timescales. As an example, the ENS monthly system is initialized and run for the first 15 days at Tco639 resolution. To run comparable re-forecasts would be very expensive (there are currently 11 members in the ENS extended reforecasts). The possibility of one seamless ENS is being considered for the future, but there are no immediate plans to do this.

The increase in computational cost is around a factor of 8.

Technical corrections:

Page 2, line 28: Remove the double "all": "... to document all all of them ...."

Page 6, line29: change "where as" to "whereas"

Page 11, line 8: change "If follows" to "It follows"

Page 16, line 2: "12 months" should be "11 months"

Page 20, line 23: change "Figures" to "Figure"

Page 32, line25: Remove the "a": "...also show a useful of skill."

Page 35, line 5: Remove the double "with": "Regions with with negative..."

Thank you very much for pointing out these errors, we have corrected them.

Page 37, line 11: change "forecast" to "forecasts"

I think you mean page 37, line 10, changing "annual range forecast" to "annual range forecasts." We have made this change.

**Anonymous Referee #2:**

General comments:

The manuscript describing ECMWF's new seasonal forecast system SEAS5 contains a comprehensive selection of evaluation metrics of model skill, biases, variability and teleconnections. It also presents in a fair manner the aspects in which the model performs better and worse than its predecessor SEAS4. This in my opinion is very useful information for the user. I think the paper is well structured, easy to follow and gives an adequate overview of the forecast system which will serve as a guiding document for many users, both scientists and non-scientists.

Thank you very much for these comments. Please see our responses to your specific comments in line below.

Specific comments:

Page 2 line 28: Remove one "all"

Thank you for spotting this, we have corrected it.

Page 5 line 17: Is there a reason for using CMIP5 and not the most recent CMIP6 forcings? Could this have an impact?

SEAS5 uses the same forcing as used in ERA5. This was not very clear in the text, so we have added a sentence of further explanation. For green house gas forcings, a seasonally varying climatology derived from MACC reanalysis for the period 2003-2011 (Inness et al., 2013) is scaled to capture the long term trend using CMIP5 historical data through 2000 and CMIP5 RCP 3-PD from then on.

The SEAS5 configuration was frozen in early 2017, so the CMIP6 future scenario forcings were not available. While we don't expect a big impact from using CMIP6 forcings, this has not yet been tested.

Page 7 line 10: Why is there a change from ERA-Interim to ECMWF operational analysis (is it superior?)

ERA-Interim reanalysis is not available in time for SEAS5 forecast initialization, so ECMWF's operational analysis is required to initialize the forecasts. This has been clarified at the beginning of section 2.3.1.

Page 8 lines 14-18: There are a few changes for the ocean/sea ice initialization during the hindcast period. Does this have an impact on skill/bias/variability?

A detailed sensitivity study on the subject of sea-ice and SST observation products has been carried out in Hirahara et al. 2016 for atmospheric and reanalyses applications. It was found that HadISSTv2 was the historical SST data set most consistent with the GHRSST OSTIA product used in operations. The combination of HadISSTv2 and OSTIA was then adopted by ERA5 and ORAS5.

Further considerations were given to the sea ice concentration (SIC) data set. It was found that the SIC product coming from HadISSTv2 was not suitable for assimilation in an ocean/sea-ice model. Instead, the reprocessed OSI-SAF SIC data set was tested and used (Zuo et al, 2018). We have updated the text with these details.

References:

Hirahara, S, Alonso-Balmaseda, M, de Boisseson, E, Hersbach, H, 2016: Sea Surface Temperature and Sea Ice Concentration for ERA5. ERA Report Series 26. https://www.ecmwf.int/en/elibrary/16555-sea-surface-temperature-and-sea-ice-concentration-era5

Zuo, H., Balmaseda, M. A., Mogensen, K., Tietsche, S., OCEAN5: the ECMWF Ocean Reanalysis System and its Real-Time analysis component, 2018, ECMWF Tech Memo 823.

Page 11 line 5: Is there a reason for not applying cross-validation for other indexes?

There are only two cases where cross validation wasn't applied: the EOF derived indices in Figures 11 and 12 and the SST anomalies in Figure 7. We have now applied cross validation. There is little visible change in the figures, and only a 0.01 change to the anomaly correlations in the caption of Figure 11.

Page 20 lines 7-9 and figure 19c: There seems to be a bit of degradation in DJF temperature skill over the Iberian peninsula as compared to SEAS4. For particular regions availability of SEAS4 forecast data could be advantageous.  Has SEAS4 operational forecast stopped or will it continue in the future?

SEAS4 operational forecasts have now stopped, after a short overlap period to allow all users time to transition to SEAS5. It is technically complex to keep old operational systems running reliably and ensure that changes in computer systems do not alter the performance of the system

Page 25 Fig 13.  Is there a reason for using normalized precipitation anomaly in the black box instead of the more standard NINO indices?

By "more standard NINO indices," we assume you mean SST anomaly in the Nino4 region. Molteni et al. (2015) demonstrated that in some regions, teleconnection patterns diagnosed as a function of precipitation anomalies are more representative of the response to anomalous heating than those diagnosed by SST anomalies, because the local SST anomaly only weakly constrains the local precipitation anomaly. We have now pointed this out in the text. The meridionally wider Nino4 box is used here to enable comparison with the original Molteni et al. (2015) study.

Page 34 line 9 and Fig 20. Is there a reason for not showing precipitation skill difference between SEAS5 and SEAS4, they are even discussed. I would suggest including it for completeness.

We did not show ACC differences in precipitation because they are very noisy, and in a global map the few areas of significant differences are not clearly visible. As an alternative to adding this, we have added maps of CRPSS and differences in CRPSS for precipitation (Figure 22). We believe they give a fairer representation of the difference in precipitation skill between the two systems. Please see the text for a discussion of the new figure.